



# Characterization of the air-sea exchanges mechanisms during a Mediterranean heavy precipitation event using realistic sea state modelling

César Sauvage [1], Cindy Lebeaupin Brossier [1], Marie-Noëlle Bouin [1,2], and Véronique Ducrocq [1]

[1]CNRM, Université de Toulouse, Météo-France, CNRS, Toulouse, France
[2]CNRS, Ifremer, IRD, UBO / Laboratoire d'Océanographie Physique et Spatiale (LOPS), UMR 6523, IUEM, Plouzané, France

**Correspondence:** César Sauvage (cesar.sauvage@meteo.fr)

**Abstract.** This study investigates the mechanisms acting at the air-sea interface during the heavy precipitation event that occurred between the 12-14 October 2016 over the north-western Mediterranean area, and that led to large amounts of rainfall (up to 300 mm in 24h) over the Hérault region (South of France). The study case was characterized by a very strong easterly to southeasterly wind at low level (>20 m.s$^{-1}$) generating a very rough sea (significant wave height up to 6 m) along the French

Riviera and the Gulf of Lion. In order to investigate the role of the waves on those air-sea exchanges during such extreme events a set of numerical experiments was designed using the Météo-France kilometer-scale AROME-France Numerical Weather Prediction model - including the WASP sea surface turbulent fluxes parametrization - and the wave model WaveWatchIII. Results from those sensitivity experiments in the forced or coupled modes showed that taking the waves generated by the model into account increases the surface roughness. Thus the easterly low-level atmospheric flow was slowed-down and the

turbulent fluxes upstream of the precipitating system were significantly modified. This modified the forecast of the heaviest precipitation, notably in term of location.

## 1 Introduction

The western Mediterranean region is regularly affected by Heavy Precipitation Events (HPE) characterized by a large amount of rainfall over a small area in a very short time which can lead to flash flood events causing severe damages and in some

cases, casualties (e.g. Delrieu et al., 2005; Llasat et al., 2013). Usually such events are generated by quasi-stationary Mesoscale Convective Systems (MCSs) developed eastwards of an upper-level trough and enhanced by an unstable low-level jet advecting moist and warm air towards the Mediterranean coasts (Nuissier et al., 2011; Duffourg et al., 2016). Several mechanisms leading to the initiation of the deep convection have been identified (Ducrocq et al., 2008, 2016), in particular the forcing by mountainous coastal regions surrounding the Mediterranean Sea which force the unstable low-level flow to lift and thus trigger

the deep convection. Besides the orographic lifting, deep convection can also be triggered by low-level wind convergence and cold pool due to precipitation evaporation.





The Mediterranean Sea still warm in autumn acts as a reservoir of moisture and heat that feeds the low-level flow by up to 40 to 60% according to the study of Duffourg and Ducrocq (2011) of ten HPEs over southwestern France. Therefore, air-sea exchanges during those events are key processes of these events. These exchanges, namely the sea surface turbulent heat, moisture and momentum fluxes can be modulated by the sea surface conditions such as the temperature (SST), sea state and wind. Indeed, previous studies have shown that fluctuations of SST can induce variations in the atmospheric low-level dynamics and stability and also in precipitation (e.g. Lebeaupin Brossier et al., 2006; Cassola et al., 2016; Stocchi and Davolio, 2017; Meroni et al., 2018; Strajnar et al., 2019). How the sea surface turbulent fluxes are formulated in the models has also a significant impact on the rainfall amounts simulated during HPE (Lebeaupin Brossier et al., 2008).

The momentum transfer from air to ocean strongly depends on the sea state. In case of wind-generated waves, the wave-induced stress represents a large fraction of the total stress that induces an enhancement of the drag airflow and so, modifies the wind profile and the near-surface dynamics (Janssen, 1989, 1991, 1992; Donelan, 1990). Analyses of *in-situ* data highlighted the strong relationship between the sea surface roughness length ($z_0$) and the wave age (Smith et al., 1992; Donelan et al., 1993; Drennan et al., 2003).

Nowadays, several bulk parametrizations of the sea surface turbulent fluxes includes this relationship between $z_0$ and the wave age. For example, the common used Coupled Ocean–Atmosphere Response Experiment (COARE) 3.0 sea surface turbulent parametrization (Fairall et al., 2003) enables to take into account the sea state in the momentum flux parametrization either with the formulation of Oost et al. (2002) or the one from Taylor and Yelland (2001) through the relationship between $z_0$ and the Charnock coefficient. Using the former formulation of Oost et al. (2002) in high-resolution numerical experiments of HPE, Thévenot et al. (2016) and Bouin et al. (2017) showed an impact on the location of precipitation when the sea state forcing is taken into account in the sea surface turbulent fluxes parametrization. Nevertheless, these formulas are known to produce too strong fluxes when strong winds (>20 m.s$^{-1}$) are encountered (Pineau-Guillou et al., 2018).

Besides directly controlling the momentum flux, surface roughness also impacts the turbulent heat fluxes and the turbulent structure and thickness of the atmospheric boundary layer (Doyle, 1995, 2002). For example, an impact on the thermodynamics structure and on the moisture transfer affecting the evolution of a convective system has been recently shown by Varlas et al. (2018) using an air-wave coupled system over the Mediterranean region.

In addition, under strong wind conditions, intense breaking of ocean surface waves, known as sea spray, occurs. The sea spray effect has a significant impact on the moisture and heat transfer at the air-sea interface that has been highlighted in several papers (e.g. Andreas, 1992; Andreas et al., 1995; Kepert et al., 1999; Bao et al., 2000, 2011; Bianco et al., 2011).

Based on this, recent studies implemented different formulations in order to better take into account the sea-state and sea spray effect on the sea surface roughness and heat and momentum fluxes during extreme events such as the Hurricane Arthur (Garg et al., 2018) or a Mediterranean tropical-like cyclone also named medicane (Rizza et al., 2018). They showed that including the surface waves effects significantly improved the simulated track, as well as the intensity and the maximum wind speed of the storm.

The sea state evolution can be retrieved from numerical wave model outputs, then used as a surface forcing of atmospheric models. However, coupled atmosphere-wave systems allow to take into account the feedback of the modified wind profile





to the wind-wave generation. Previous studies have demonstrated the importance of wind-wave coupling showing significant effects on the representation on the atmospheric low-level dynamics (e.g. Renault et al., 2012; Ricchi et al., 2016; Katsafados et al., 2016; Wahle et al., 2017; Varlas et al., 2018). They have shown impacts on the drag coefficient over rough sea and on the momentum flux resulting in a reduced simulated surface wind speed. Also, all these studies reassessed clearly the need of

5 a better representation of the sea state in the understanding of air-sea exchanges. The present study investigates the impact of waves on a Mediterranean HPE using a kilometric-scale coupled atmosphere-wave system. The coupling involves the wave model WaveWatchIII and the AROME Numerical Weather Prediction model with a new fluxes parametrization (WASP) that takes explicit wave parameters to compute the wave age, then $z_0$ and the momentum and heat fluxes. The Heavy Precipitation Event studied here occurred during 12 to 14 October 2016 with two main convective areas, one over sea and one hitting the

10 south of France (Hérault). Strong wind conditions and a very roughness sea were observed during this event, making it a well suited case to study the influence of waves on the atmospheric low levels and heavy precipitation systems.

A detailed description of the experimental protocol and of the fluxes parametrization is given in section 2. The validation of the reference experiments against available atmospheric and wave observations is done in section 3. Then, in section 4, a description of the event divided into separate phases is given. Section 5 presents the results of the sensitivity analysis done by

15 comparing our different numerical simulations. Conclusions and discussions are finally given in section 6.

## 2 Numerical Set-Up

### 2.1 The atmospheric model

The non-hydrostatic AROME Numerical Weather Prediction (NWP) model (Seity et al., 2011) is used in this study. The AROME configuration used here is the one operationally used at Météo-France with a 1.3 km horizontal resolution and a do-

20 main centered over France which covers notably our area of interest, the North-Western Mediterranean Sea (Fig. 1a). The vertical grid has 90 hybrid $\eta$-levels with a first level thickness of almost 5m. The time step is 50 s. In AROME, the advection scheme is semi-Lagrangian and the temporal scheme is semi-implicit. The 1.5 order TKE turbulent scheme from Cuxart et al. (2000) is used. Thanks to its high resolution, the deep convection is explicitly solved in AROME while the shallow convection is solved with the Eddy Diffusion Kain Fritsch (EDKF, Kain and Fritsch, 1990) parametrization. The ICE3 one-moment microphysical

scheme (Pinty and Jabouille, 1998) is used to compute the evolution of five hydrometeor species (rain, snow, graupel, cloud ice and cloud liquid water). The surface exchanges are computed by the SURFace EXternalisé (SURFEX) surface model (Masson et al., 2013) that considers four different surface types: land, towns, sea and inland waters (lakes and rivers). Exchanges over land are computed through the ISBA (Interaction between Soil, Biosphere and Atmosphere) parametrization (Noilhan and Planton, 1989). The formulation from Charnock (1955) is used for inland waters whereas the Town Energy Balance (TEB)

scheme is activated over urban surfaces (Masson, 2000). The treatment of the sea surface exchanges in AROME-SURFEX is detailed below (sections 2.3.1 and 2.3.2). Output fluxes are weight-averaged inside each grid box according to the fraction of each respective tile, before being provided to the atmospheric model at every time step.



## 2.2 The wave model

The wave model is WaveWatchIII (hereafter WW3, version 5.16, The WAVEWATCH III Development Group 2016; Tolman 1992). The model domain covers the North-Western Mediterranean Sea at 1/72° horizontal resolution (Fig. 1b). The bathymetry is coming from the NEMO-NWMED72 ocean model configuration described in Sauvage et al. (2018b), itself originally built

from the interpolation of a 1/120° horizontal resolution topography with a dedicated treatment for islands, coastlines and river mouths delineation.

The set of parametrizations from Ardhuin et al. (2010) is used like for most of the wave forecasting centers (Ardhuin et al., 2019). Thus, the swell dissipation is computed with the Ardhuin et al. (2009) scheme and the wind input parametrization is adapted from Janssen (1991) with the adjustments of Bidlot et al., 2005, 2007. Nonlinear wave-wave interactions are computed

using the discrete interaction approximation (DIA, Hasselmann et al., 1985). The parametrization of the reflection by shorelines is described in Ardhuin and Roland (2012). Moreover, the computation of the depth-induced breaking is based on Battjes and Janssen (1978)'s algorithm and the bottom friction formulation follows Ardhuin et al. (2003).

## 2.3 Atmosphere-wave coupling

### 2.3.1 Bulk iterative equations

The sea surface turbulent fluxes are calculated through the bulk formulae described as follows:

$$\tau = \rho C_D \Delta U^2, \tag{1}$$

$$LE = \rho L_v C_E \Delta U \Delta q, \tag{2}$$

$$H = \rho c_{pa} C_H \Delta U \Delta \theta, \tag{3}$$

with $\rho$ the air density, $c_{pa}$ the air heat capacity and $L_v$ the vaporization heat constant. $\Delta U$, $\Delta q$ and $\Delta \theta$ represent the air-sea

gradients of velocity, specific humidity and potential temperature near the surface, respectively. $C_D$, $C_E$ and $C_H$ represent the transfer coefficients. Each transfer coefficient can be defined such as:

$$C_X = c_x^{1/2} c_d^{1/2}, \tag{4}$$

where $x$ being $d$ for wind speed, $\theta$ for potential temperature and $q$ for water vapor humidity. Therefore:

$$c_x^{1/2}(\zeta) = \frac{c_{xn}^{1/2}}{1 - \frac{c_{xn}^{1/2}}{\kappa}\psi_x(\zeta)} \tag{5}$$

and:

$$c_{xn}^{1/2} = \frac{\kappa}{ln(z/z_{0x})} \tag{6}$$

with the subscript $n$ refering to neutral ($\zeta = 0$) stability, $z$ to the reference height and $\psi_x$ is an empirical function describing the stability dependence of the mean profile, $\kappa$ is von Karman's constant.





The sea surface roughness length $z_0$ is defined by the Charnock's equation (Charnock, 1955):

$$z_0 = \frac{\alpha_{ch}.u_*^2}{g} + \frac{0.11.\nu}{u_*}, \tag{7}$$

with $\nu$ the kinematic viscosity of dry air, the friction velocity $u_*$ and the Charnock coefficient $\alpha_{ch}$.

### 2.3.2 Wave impact on the Charnock coefficient

One common method to couple developing waves and stress is to make the Charnock coefficient $\alpha_{ch}$ explicitly dependent on the sea state by computing it either in the wave model from the wave spectra (e.g. Janssen et al., 2001) or in the atmospheric model as a function of the wave age (Mahrt et al., 2001; Oost et al., 2002; Moon et al., 2004). Studies based on observations (e.g. Oost et al., 2002) express it as a power function:

$$\alpha_{ch} = A.\chi^{-B}, \tag{8}$$

with coefficients A and B either constant or depending on the surface wind speed, and the wave age $\chi$ defined as $\chi = \frac{c_p}{U_a}$ with $c_p$ the peak phase speed and $U_a$ the near surface wind. Sea state can mainly be defined using the wave age. Wind sea corresponds to waves, generated by local wind, which are still growing (wave age < 0.8) or in equilibrium with the wind (wave age between 0.8 and 1.2), and are aligned with the local wind. These waves (and only these ones) benefit from momentum transfer from the atmosphere to grow and, as such, are coupled with the wind. Conversely, swell (that does not depend on the momentum from the atmosphere) corresponds to waves generated by a remote or past wind field, and are characterized by wave-age above 1.2 or are not aligned with the local wind.

Assuming that the water depth is infinite (in practice, as soon as the depth is much larger than the dominant waves), the phase speed of the waves can be expressed as:

$$c_p = \frac{gT_p}{2\pi}, \tag{9}$$

with $T_p$ the peak period of the waves and $g$ the acceleration of gravity.

Keeping the coefficients A and B constant with wind speed results in drag coefficient and wind stress too strong in strong wind conditions (wind speed above 20 m.s$^{-1}$) as shown by Pineau-Guillou et al. (2018). In order to tackle this, and to reproduce the saturation or the decrease of the drag coefficient observed in strong to cyclonic winds (e.g. Powell et al., 2003), we take advantage of a new parameterization called WASP (Wage-Age Stress dependant Parametrization). This approach considers that the wind-speed range where the wind stress transferred to the sea surface mainly sustains the wave development (through interaction with non breaking waves) is 5 to 20 m.s$^{-1}$. Above 20 m.s$^{-1}$, the contribution of breaking waves to the wave stress is dominant and the wave age is not an appropriate parameter to represent the sea state effect on the surface roughness. Below 5 m.s$^{-1}$, by very weak wind, the Charnock parameter is mainly controlled by the viscous term (second term in the right-hand side, Eq. 7). In order to reproduce these different mechanisms and the decrease of the drag coefficient by very strong wind, the Charnock parameter of the WASP parameterization is piecewise continuously defined following Eq. 8, coefficients A and





B being polynomial functions of the surface wind speed. In weak to strong wind regimes where wind stress observations are numerous and consistent with each other (*i.e.* until 23 m.s$^{-1}$), it has been fitted to datasets used to build the COARE 3.5 parameterization (Edson et al., 2013). The temperature and humidity roughness lengths $z_{0T}$ and $z_{0q}$ (see Eq. 6) that define the corresponding neutral transfer coefficients have been adjusted for the resulting sensible and latent heat fluxes to match the

COARE 3.0 (Fairall et al., 2003) parametrization for surface wind speeds up to 45 m.s$^{-1}$. The stability functions $\psi_x$ (Eq. 5) are a blend of the Kansas-type functions (Businger et al., 1971) with a profile matching the asymptotic convective limit (Fairall et al., 1996).

### 2.3.3 Coupling

The coupling is performed using the SURFEX-OASIS coupling interface (Voldoire et al., 2017) that manages the exchanges

between the AROME and WW3 models. AROME-SURFEX provides the two components of the near surface wind speed to WW3 whereas WW3 provides $T_p$ to AROME-SURFEX. The OASIS coupler (Craig et al., 2017) allows to choose the coupling frequencies and the interpolation methods for the exchanged fields.

Also, as the AROME domain is larger than the WW3 domain, in particular covering also a part of the Atlantic Ocean, there is no air-wave coupling for the marine zones not covered by WW3 (i.e. for the grey zones in Fig. 1a) and, there, $\chi$ is directly

estimated in WASP as a function of wind with $T_p = 0.5 \times U_a$ and thus $c_p = \dfrac{g U_a}{4\pi}$.

### 2.4   Set of simulations

In this study, three kinds of simulations were examined: wave-only, atmosphere-only and wave-atmosphere coupled simulations.

First a wave-only simulation (named WY) was run. For this, WW3 runs from the 5 to the 15 October 2016 with initial

condition ($H_s$,$T_p$) set to 0 and using a near-surface wind forcing coming from AROME forecasts at a hourly frequency (+1h to +24h each day, see Sauvage et al. (2018a)). The period from 5 to 12 October served here as a spin-up period and will not be considered in the following. Boundary conditions for the wave model consists in 8 points spectra distributed along the domain and provided by a WW3 global simulation running at 1/2 ° resolution run at IFREMER (Rascle and Ardhuin, 2013).

Atmosphere-only simulations were done with AROME. Each AROME simulation was composed of forecast runs that start

every day (12, 13 and 14 October 2016) at 00UTC from AROME operational analyses and that last 42h. Hourly boundary conditions came from the ARPEGE (Action de Recherche Petite Echelle Grande Echelle, Courtier et al. (1991)) operational forecasts, except the Sea Surface Temperature (SST) that came from the global daily analysis of Mercator Océan International (1/12°-resolution PSY4 system, Lellouche et al., 2013). The first atmosphere-only (AY) simulation was done without any wave information, meaning that $\chi$ was set as a function of the near surface wind with $T_p = 0.5 \times U_a$. The second atmosphere-only

run (AWF) was forced by the hourly $T_p$ from the WY simulation.

Finally, a two-way coupled AROME-WW3 simulation (AWC) was done following the description given in section 2.3.3. The SST field, the atmospheric initial and boundary conditions of AWC were the same as in the atmosphere-only simulations. Also, the wave boundary conditions in AWC were the same as for WY. Wave initial conditions come from restart files, first





from WY for the forecast starting on 12 October 00UTC, then for the following days, from the previous AWC forecast run (after 24h). The coupling frequency was set to 1h in both ways. Each exchanged field was interpolated with a bilinear method.

## 3 Validation of the experiments

### 3.1 Available observations

In order to validate the simulations, we collected several observations of the surface and near surface in the north-western Mediterranean area (see Fig. 1b that displays the observations over the WW3 domain).

   Data from 14 moored buoys (listed in Tab. 1 and plotted in Fig. 1b) available either from Copernicus Marine Environment Monitoring Service (CMEMS, http://marine.copernicus.eu/) database or from the HyMeX program database (http://mistrals. sedoo.fr/HyMeX/) were first used for validation. These platforms measure a wide variety of near-surface variables that may

include sea temperature, salinity, wave parameters like the significant wave height ($H_s$) and peak period ($T_p$), but also some atmospheric parameters such as the 2m-air temperature, relative humidity, the 10m-wind speed, direction and gusts. Added to those buoys, 6 coastal surface weather stations from the Météo-France network were used mainly around the Gulf of Lion and in Corsica in order to complete the coverage over the area of interest concerning atmospheric *in-situ* data.

Altimetric data from two satellites crossing the area during the event were used for the significant wave height validation. The

15 first satellite is Jason-2 (OSTM/Jason-2 Products Handbook , 2008) from the joint CNES/NASA oceanography mission Jason. Altimetric measurements used are the Geophysical Data Record (GDR) from the MLE4 (Maximum Likelihood Estimator) altimeters retracking algorithm corrected following a buoy comparison method: $H_s\_cor = 1.0149 \times H_s + 0.0277$. The second dataset was obtained from GDR data of the SARAL/AltiKa satellite (SARAL/AltiKa Products handbook, 2013) that also uses the MLE4 altimeters retracking algorithm but simply removing erroneous $H_s$ using a threshold relationship. Both satellites

combined gather 292 measures of $H_s$ during the period between 12 and 14 October 2016.

   To validate the rainfall accumulation the ANTILOPE product from Météo-France were used (Laurantin, 2008). This product merges rain gauges and radar data. This analysis was complemented by the use of the Météo-France radar composite images over western Europe.

### 3.2 Validation of AWF and WY

For the validation of our wave and atmospheric reference simulations, WY and AWF respectively, the time-series from 12 to 14 October were built using the first 24h of simulation starting each day (Fig. 2). Observations and simulations were compared using the nearest grid point in space and time from the model (WW3 or AROME) to the observation ones. The bias, the root mean square error (RMSE) and the correlation were computed and summarized in Table 2.

   During the entire event, the sea state appeared to be well represented by WY with correlation of 0.90 for $H_s$ and $T_p$. Looking

at Figure 2, $H_s$ and $T_p$ seemed to be underestimated during the event. This was confirmed by negatives biais of -23 cm for





$H_s$ and -0.79 s for $T_p$, as by the comparison of the simulated $H_s$ against satellite data with an average biais of about -0.17 cm (Table 2). Yet, a good correlation of 0.78 was obtained with satellite data.

In AWF, the wind speed and direction were quite well represented with a correlation of 0.64 and 0.86, respectively (Table 2). They were very slightly overestimated during the event with an averaged bias of 0.04 m.s$^{-1}$ and 2°. Looking the wind speed
correlation over different regions, *i.e.* Gulf of Lion and Balearic Sea, some differences can be highlighted. In the Gulf of Lion and along the French Riviera a correlation coefficient of 0.82 was found. Looking at the buoys located in the Balearic Sea, *i.e.* where the wind was weaker, the simulation well represents the wind speed value but a low correlation (0.45) is found.

By looking more in detail at the French western coastal buoys, such as Leucate (Fig. 2) located on the most western part of the Gulf of Lion, a large underestimation of the wind speed was found. The observed wind speed reached several times values
between 18 and 20 m.s$^{-1}$ whereas the simulation reached only 15 m.s$^{-1}$. Added to this, at Sète, the wind intensity was in good agreement but decreased faster than observed (not shown). It also can be noticed a delay at the end of the event between the simulation and observations, looking at the Lion buoy (Fig. 2) when a transition to a northerly wind occurred.

The validation against *in-situ* data for the 2m-air temperature (T2M) and relative humidity (RH)( Tab. 2) showed that AWF represented quite well these two atmospheric parameters with correlation coefficients of 0.68 and 0.78, respectively, despite
small overestimations (averaged biases 0.26°C and 2.23%, respectively). These scores, together with the validation against low-level wind and wave parameters, permitted to be confident in the use of AWF to investigate the evolution of the turbulent fluxes at the air-sea interface during the HPE.

## 4   Event description

The studied HPE occurred the 13 and 14 October 2016 over the north-western Mediterranean Sea. This event could be defined
as a typical "Cyclonic Southerly" (CS) case, following the four synoptic type classification of Nuissier et al. (2011). Indeed, the synoptic meteorological situation (Fig. 3) was characterized by a trough in altitude extending from the British Islands to Spain and associated with a lowering of the tropopause (Fig. 3a) inducing a south-westerly flow over south-eastern France. A cyclonic circulation took place at low levels (Fig. 3b) with a high moisture content over the Gulf of Lion and a strong south-easterly flow that originated from south-eastern Tunisia. During the night and the following day, the trough moved eastwards from the
Bay of Biscay to the Gulf of Lion along with cold and warm fronts (Fig. 3c) and the low level flow shifted also eastwards.

It was characterized by four periods that can be distinguished using observations over land and the reference simulation (AWF) for the marine low-level conditions: (I) initiation stage, (II) mature systems, (III) north-eastward propagation and (IV) Tramontane wind onset. In the following, a detailed description of the chronology of the event and the mechanisms involved is done. In that purpose, the 42h forecast starting on the 13 October 2016 00UTC of AWF was used along with observations.

### 4.1   Chronology of the convective systems

The Phase I, from 03UTC to 18UTC on 13 October 2016 (+3h to +18h in the simulation), was marked by the triggering of deep convection and the stationarity of the two main systems. The first deep convective system was triggered at the Cévennes





foothills, south of the Massif Central, (Figs. 1, 4a and 5a), where the unstable rapid south-easterly marine flow encountered orography (Fig. 5c). The second one was a MCS associated with large precipitation over sea (Figs. 4a and 5a). It formed at the convergence between the warm south-easterly flow, associated with high CAPE values, with the colder and drier easterly flow from the Alps and Ligurian Sea (Figs. 5c,d). These two convective systems were well represented in the reference simulation

(Figs. 4a and 5a) in terms of location and rainfall amounts compared to observations. The simulated radar reflectivities corresponded also quite well to observations (not shown) except in the northeastern Spain where too active convective systems were simulated.

During Phase II, from 19UTC on 13 October to 03UTC on 14 October (+19h to +27h in the simulation), the precipitating system over the Hérault region stayed stationary. Its intensity increased in the observation (Fig. 4b) whereas precipitation totals

for the second system over sea decreased. In AWF the simulated precipitating system over sea started to shift towards east while precipitation over the Hérault region decreased (Fig. 6a). The simulated cold front progresses eastwards earlier than in the observations and thus, the southern flow started also to shift (Fig. 6d) as for the convergence line over sea (Fig. 6c).

The Phase III, from 04UTC to 10UTC on 14 October (+28h to +34h in the simulation), was marked by the north-eastward propagation of the system. As the western front was moving eastwards, the system over the Hérault region shifted toward the

French Riviera (Fig. 7c) leading locally to a decrease in precipitation total (< 60 mm over the Hérault region, Fig. 4c). In the simulation, the system over sea also moved eastwards, extended from West of Corsica to the French Riviera along with a decrease in the rainfall amounts (Fig. 7a). It appeared that the simulated eastward propagation was earlier than observed. Consistently, the southerly flow was also shifted and was located between Corsica-Sardinia and the continental Italy (Fig. 7d) with main convergence area over the Ligurian Sea. At the same time the intensity of the inland system (over the Hérault region)

was overestimated compared to observations (Figs. 4c and 7a).

The last Phase (IV), from 11UTC on 14 October (+35h to the end of simulation, not shown), was characterized by the end of precipitation over France (Fig. 4d) and the beginning of a new wind regime in the Gulf of Lion, with a dry and cold north-westerly flow corresponding to the regional Tramontane wind regime. The warm southern flow was thus limited to the South-East of the simulation domain and fed the precipitating system located from North of Sardinia to continental Italy region

(Fig. 4d). For this period, the simulation was in good agreement with observations in term of rainfall location and amounts.

## 4.2 Evolution of the sea state

Three different areas can be distinguished in our domain, mainly represented by the three moored buoys (Fig. 1, Tab. 1).

– The Tarragona buoy, where the wind was weak, was situated in a long fetch area. There was swell all along the event, first aligned with the southeasterly wind in Phase I and then crossed, as wind and waves were opposite.

– The Lion buoy was located where the easterly wind was stronger during Phase I generating a young wind-sea with strong $H_s$. It evolved to a well-developed wind-sea during Phase II and then to a swell as the fetch became longer in this area.

– The Azur buoy was located in the strong easterly wind all along the event. Characterized by a short fetch, a wind-sea was continuously produced in this area.





During Phase I, a strong easterly wind (between 15 and 20 m.s$^{-1}$, Figs. 5b and 2) affected the Ligurian Sea, the French Riviera till the Gulf of Lion. This created a "wind sea", with young waves (see Azur buoy on Figs. 8 and 5e) aligned with the wind and associated with moderate $H_s$ (<3 m, Fig. 5f). In fact, all along the event the waves in this area were directly generated by the local wind (Fig. 8) with low values of wave age (< 1) and similar direction between wind and waves. In the Gulf of

5 Lion a rough to very rough sea was observed ($H_s \sim$ 4-5 m) associated with $T_p$ about 8 s (Fig. 2). Still, weak wage age (Figs. 5e and 8) were found here due to strong winds. Thus, in this area the waves were mainly generated by the wind above. In the Balearic Sea high wave age were simulated (> 1.2, Fig. 8) under a weak south-westerly flow (Fig. 2) inducing weaker $H_s$, corresponding to moderate sea. The sea state in this area corresponded to a swell coming from the south.

During Phase II, $T_p$ was maximum ($\sim$ 10 s observed) in the Gulf of Lion as for $H_s$, which reached about 6 m corresponding

to a very rough sea which is rather exceptional in the Mediterranean Sea. This well-developed wind-sea (wave age between 0.8 and 1.2) was due to the continuously easterly wind blowing since the first phase (Fig. 6b and 2). The intensity of the easterly flow decreased in the Gulf of Lion but increased in the Ligurian Sea (+/- 2-3 m.s$^{-1}$, Figs. 6b and 2). In the Balearic Sea, the wave age were still high but with wave direction changing progressively from easterly to north-easterly. This corresponded to the swell coming from the Ligurian Sea as a weak south to north-westerly wind (< 10 m.s$^{-1}$) was blowing locally (Figs. 6b,e,

8 and 2).

During Phase III, $H_s$ started to decrease over the domain (Figs. 7f and 2) and the observed $T_p$ was maximum in the Balearic Sea and in the western part of the Gulf of Lion, associated with swell (Figs. 7e, 2 and 8). As the front was moving eastwards, a significant decrease in the easterly wind was observed (up to -7m.s$^{-1}$, Figs. 7b and 2), whereas the wind speed was the strongest in the Gulf of Genoa. Indeed, the wind in the Gulf of Lion started to change direction with a transition to a northerly

wind (see Leucate, Sète and Lion in Figure 2). As in the Balearic Sea (Fig. 7e and 8) the wind was changing direction from north-westerly to westerly.

Finally, during Phase IV, $H_s$ kept decreasing over the domain (<3 m, Fig. 2). $T_p$ significantly decreased in the Gulf of Lion whereas the highest values were still located in the Balearic Sea (Fig. 2). In both locations, the swell generated along the French Riviera was present (Fig. 8) while the easterly wind significantly decreased (now <14 m.s$^{-1}$, Fig. 2). A north-westerly wind

(Tramontane) onset in the Gulf of Lion was observed (Fig. 2), while a westerly wind was blowing over the Balearic Sea.

### 4.3 Air-sea interface

The latent heat flux (LE) was quite low over the domain, as displayed by the Figure 5g. During Phase I, the cold and dry air from the Alps became rapidly warmer and more humid as it flowed westwards over sea. The evaporation started in this area and marked also the location of the largest values of LE (over 300 W.m$^{-2}$, Fig. 5g). Added to this, this region was characterized by

30 the strongest humidity transport, due to the easterly flow, towards the Gulf of Lion as the air humidity increased (over 94%). Under the strong easterly wind, along the French Riviera, the difference between SST and T2M was up to 4°C (5°C locally) and thus with high values of sensible heat flux (H) (over 150 W.m$^{-2}$, Fig. 5g). The warm (>23°C) and humid (over 85 %) southern flow did not produce large heat fluxes (Fig. 5g). It can be noticed that in the Balearic Sea there was warm and dry air masses but the weak southwesterly wind blowing in this area limited evaporation and heat fluxes. The momentum flux was the





largest under the strong easterly wind in the Gulf of Lion up to 1.5 N.m$^{-2}$ and locally up to 1.2 N.m$^{-2}$ under the southeasterly flow (Fig. 5h). It remained lower than 0.3 N.m$^{-2}$ in the rest of the domain.

As the system moved eastwards during Phase II, the rapid low-level flow moved from the Gulf of Lion to the Ligurian Sea. The evaporation kept increasing in this area and the dry air in the Gulf of Genoa became nearly saturated in the Gulf of Lion. The largest values of momentum flux were at this time located along the French Riviera (Fig. 6h). LE decreased in the Gulf of Lion by more than 50 W.m$^{-2}$ and increased along the French Riviera and the Gulf of Genoa to reach more than 360 W.m$^{-2}$ (Fig. 6g). As the cold front was shifting, the low-level air mass in the Balearic becomes drier and it pushed the humid southerly flow to the east. Maximum values of H are also shifted in the Gulf of Genoa to reach 200 W.m$^{-2}$ (Fig. 6g) whereas H significantly decreased in the Gulf of Lion.

During Phase III, drier air (RH <70%) was now located from the Balearic Sea to the coast of Sardinia. In the Gulf of Lion, low-level air kept getting drier whereas moist air was now mainly located along the French Riviera and in the Gulf of Genoa where precipitation occurred. Under precipitation, H increased to 250 W.m$^{-2}$ (Fig. 7g). LE significantly decreased by 100 W.m$^{-2}$ along the French Riviera but was still maximum in the Gulf of Genoa (Fig. 7g). A large decrease was also noticed in the momentum flux (by 1 N.m$^{-2}$) along the French Riviera and maximum values were found in the Gulf of Genoa (Fig. 7h).

During the last Phase (IV), with the large decrease in the wind intensity and the precipitation now located over Italy, RH decreased along the French Riviera and the Gulf of Genoa associated with a large decrease in the heat fluxes (by 150 W.m$^2$). The momentum flux was now lower than 0.3 N.m$^{-2}$ in the Gulf of Genoa.

A rapid analysis of the relationship between the heat fluxes (H and LE) and atmospheric parameters ($U_a$, temperature gradient, humidity gradient) using scatterplots was done (not shown). It highlighted that the sensible heat flux was more correlated to the temperature gradient at the air-sea interface (0.56), related to cold air present in the easterly flow or below precipitation. Conversely, the latent heat flux was more correlated to the wind (0.49) than to the gradient of humidity (0.31). In summary, the maximum turbulent fluxes were associated with the easterly low-level jet due to strong winds and large air-sea temperature (moisture) gradients for heat fluxes. In the southerly flow, heatfluxes appeared more limited despite moderate wind. This highlighted the role of the Ligurian easterly flow in extracting heat and moisture from sea and in providing them to the MCSs.

Finally, looking at the different phases of the event, some areas emerged as potential regions where the waves should have an impact on the low-level flow. Indeed, during Phases I and II, an effect on the momentum flux was expected on areas of strong easterly wind and of wind-sea, especially over the French Riviera and the Gulf of Lion in Phase I. Those regions were also the places with the highest heat fluxes during the event and thus, were more likely to be affected by the sea state. So, the sensitivity to the impact of the representation of sea state will be particularly investigated these areas in the following.

## 5 Sensitivity analysis

In this section, the goal is to better understand the impact of the waves on the sea surface turbulent fluxes and to evaluate the impact on the HPE forecast. We focused on Phases I and II, on 13 October at 14UTC and on 14 October 2016 at 00UTC




(corresponding to +14h and +24h of forecast, respectively, starting on 13 October 00UTC) as differences between the three simulations AY, AWF and AWC were well established.

## 5.1 Low-level flow

### 5.1.1 Impact of the waves: AWF versus AY

In the following, AWF, which takes into account the sea state, is compared to AY the atmosphere-only simulation (see section 2.4). Figures 9a,c present the difference of the sea surface roughness length ($z_0$) between AWF and AY. During the Phase I, compared to AY, an increase of $z_0$ in AWF was found over the wind sea under the strong easterly wind in the Gulf of Lion and along the French Riviera (from $2 \cdot 10^{-3}$ to $4 \cdot 10^{-3}$ m). In the rest of the domain much lesser $z_0$ differences were noticed (less than $2 \cdot 10^{-4}$ m). These changes induced an increase of the drag coefficient $C_d$ up to $0.8 \cdot 10^{-3}$ and led to an increase of more

than 0.1 N.m$^2$ of the momentum flux. Finally, it resulted in a decrease of the 10m wind speed intensity of the strong easterly flow by more than 1 m.s$^{-1}$ over a large area between the Gulf of Lion and the French Riviera (Fig. 10a). During the Phase II, along the French Riviera and the Gulf of Genoa characterized by the strong easterly wind and a young wind sea (Figs. 6b,e), $z_0$ increased by more than $2 \cdot 10^{-3}$ m and up to $1 \cdot 10^{-2}$ m in AWF compared to AY (Fig. 9c). Knowing that in AY $z_0$ barely goes over $3 \cdot 10^{-3}$ m these differences correspond to an increase of more than 100% of the values in AY. Under the convective

system some difference dipoles were found. In the Gulf of Lion a slight increase of $z_0$ was seen. Whereas the differences in $z_0$ in Liguria were seen from the beginning of the simulation, differences under the MCS and in the Gulf of Lion appeared to be more likely resulting from differences in the movement of the convective system over sea induced by the decrease of the wind intensity during Phase I. By the same mechanisms as in Phase I, the increase of $z_0$ upstream of the MCS (*i.e.* along the French Riviera) directly impacted the $C_d$ which increases in AWF by $0.2 \cdot 10^{-3}$ to locally more than $1 \cdot 10^{-3}$. This led to an

increase of the wind stress in this area between 0.1 and 0.3 N.m$^2$ and resulted in a slow down of the 10-m wind speed along the French Riviera between 1 to 2 m.s$^{-1}$ (Fig. 10c). Larger differences were found under the convective system, but appeared inhomogeneous in space and time. Thus, the results confirmed the primary effect of the representation of sea state as notably highlighted by Thévenot et al. (2016) and Bouin et al. (2017), *i.e.* an increased surface roughness and wind stress when sea state is taken into account that slow down the upstream low-level flow. In the two sub-areas delineated in Figure 10c, it was found

that on average the slow down of the 10-m wind speed was obtained in both areas during the four Phases. More especially in the Gulf of Lion during the Phase I, an averaged slow down of 0.9 m.s$^{-1}$ was noticed. This represented a decrease of about 6% of the average wind intensity in AWF. The same result was found during Phase II along the French Riviera with an averaged decrease of 0.9 m.s$^{-1}$ (-7%). Scores did not appear to be significantly changed between AWF and AY (Table 2). However, a lower bias in the wind intensity was found (0.04 m.s$^{-1}$ in AWF against 0.22 m.s$^{-1}$ in AY) and was actually mostly due to a

large improvement at the Azur buoy where the bias was reduced from 0.42 m.s$^{-1}$ in AY to 0.08 m.s$^{-1}$ in AWF.

Figures 11a,c present the heat fluxes (latent and sensible fluxes, resp.) differences between AWF and AY. Along the French Riviera, where the latent heat flux was the strongest, a decrease was obtained in AWF during Phases I and II. However, this corresponded to a small decrease (5 W.m$^{-2}$ on average) equivalent to 2% of the total averaged latent flux. Relatively larger





differences, positive and negative, were found under the convective system. But on average these differences were small, representing +/- 2% (3 W.m$^{-2}$). They were very likely related to differences in term of intensity of the convection within the MCS and its location and not to a direct effect of the waves. Differences in the sensible heat flux (Fig. 11c) were mainly located under the precipitation with very weak differences along the French Riviera.

### 5.1.2 Impact of the coupled system: AWF versus AWC

Figures 9b,d present the differences in $z_0$ between AWF and the atmosphere-wave coupled system AWC. During Phase I, $z_0$ in AWC was increased up to $2 \cdot 10^{-3}$ m over the French Riviera and the eastern part of the Gulf of Lion (Fig. 9b). As a result a slight decrease of the 10m wind speed intensity was found, about 0.6 m.s$^{-1}$ (Fig. 10b). During Phase II, smaller differences were obtained along the French Riviera. It corresponded to a small increase of $z_0$ in AWC of about $1 \cdot 10^{-3}$ m, under the strong easterly wind (Fig. 9d). The 10m-wind speed was decreased in AWC by no more than 0.3 m.s$^{-1}$ (Fig. 10d). The smaller impact on the low level dynamics in AWC can be explained by the feedback of the wind on the waves. Indeed, it was found that on average during the event $H_s$ was decreased in AWC by 12% and $T_p$ by 7%. Since, in WY we had already an underestimation of the wave parameters, this decrease in AWC induced larger biases (Table 2). Larger differences in the wind intensity were still found under the convective system and downstream of it in the Gulf of Lion (Fig. 10d). However, those were not really consistent time-to-time and were mainly due to the movement of the system and the slow down of the wind during Phase I.

Figures 11b,d illustrate the differences in the heat fluxes. Either for LE or H very small variations were noticed, less than 10 W.m$^{-2}$ along the French Riviera. As before, the larger differences in the Gulf of Lion were more likely to be induced by the movement and the convective cells evolution of the MCS.

Thus, in average, coupling showed here only minor effects on the dynamics and on the heat and moisture exchanges below the upstream low level flow. One main explanation for this small effect might be that the waves used in AWF and in AWC were really close to each other in term of spatial and temporal resolution, both simulated using WW3. Locally, effects on the dynamics can be significant especially in strong wind and wind-sea areas where we found a decrease of the wind speed and of $H_s$ and $T_p$.

### 5.2 Precipitation

The maximum peak rainfall amounts in 24h simulated over the Hérault region were 273 mm in AY, 278 mm in AWF and 271 mm in AWC and agree with the ANTILOPE maximum value of 287 mm. Larger differences were found for the convective system over sea, with maximum peak rainfall amount in 24h of 348 mm in ANTILOPE, but only 214 mm in AY, 187 mm in AWF and 188 mm in AWC. Note, however, that the ANTILOPE rainfall amount estimations over sea were not corrected with rain gauges and might contain some inaccuracies due to the distance from the ground-based radars.

Figure 12 presents the differences in the 6h-rainfall amount between 13 October 18UT and 14 October 00UTC. On average, the total amount of rainfall in the sub-areas in Figure 12a, corresponding to the MCSs locations, was about the same in all simulations. A displacement of approximately 40 km eastwards of the precipitation over sea was found however in AWF compared to AY (Fig. 12a). This displacement was directly related to the decrease of the wind speed along the French Riviera





(Fig. 10c) and thus to the convergence line that was located more east. For the convective system over the Hérault area only a slight shift (few km) of the maximum peak was seen. Also, in this area the simulated precipitation amounts in AY and AWF were both too far inland (Fig. 4b). In AWC (Fig. 12b) a slight shift of few kilometers westwards was obtained when compared to AWF.

Thus, these differences in the precipitation forecasts highlighted here the indirect effects of taking sea state into account: first, a modification of the position of the convergence line at sea related to the speed of the low-level easterly flow and then a small modulation of the intensity of the associated convection likely due to differences in term of heat fluxes upstream over the Ligurian Sea. These differences, which concern the MCS at sea, then induced low-level flow disturbances downstream in the Gulf of Lion, but with relatively little impacts on the dynamics of the precipitating system that affected the Hérault area. This demonstrated that the mechanism involved in the formation of this inland system, *i.e.* the orographic uplift, then the reinforcement by the convergence between the southerly flow and the large-scale front, were dominant features and appear, for that precipitating system, less sensitive to the sea surface conditions.

## 6    Conclusions

The Mediterranean HPEs are known to be violent events and are quite often associated with strong wind conditions and thus very rough sea state. This study investigated the role of the representation of the sea state during the HPE that occurred between the 12 and 14 October 2016 south of France. Thanks to sensitivities experiments, the strong air-sea interactions during the event were analysed and allowed us to evaluate the impact of the representation of the sea state in the forecasting system.

In this purpose, a set of high resolution (1.3 km) numerical simulations was realized using the atmospheric model AROME and the wave model WW3, both in stand-alone mode or in the two-way coupled atmosphere-wave mode. To describe the turbulent fluxes that control the sea surface exchanges the innovative parametrization WASP was used as it is particularly designed to be used in coupled way with a wave model and allows to directly take into account the peak period $T_p$ in the calculation of the surface roughness length $z_0$.

Using observations and the reference simulation (AWF), we highlighted that the studied event was characterized by a convergence between a warm and moist southerly flow with a dry and cold easterly flow, that triggered convection over sea. A second convective system, south of France, was initiated by an orographic uplift and was fed by the easterly flow. Both systems produced a large amount of precipitation. Three characteristic regions emerged from the analysis. First, the Balearic region was affected by weak wind and swell all along the event. Then, the Gulf of Lion was located at first where the easterly flow was maximum producing a young sea with high $H_s$ and strong air-sea fluxes. As the system was moving eastwards with the maximum wind intensity, the sea state evolved from a well-developed sea to a swell in this region. Finally, the French Riviera, was affected all the event by the strong easterly wind generating a wind sea. There the heat fluxes were the most intense.

The simulation results were compared to various observations such as moored buoys for atmospheric and waves parameters (completed with Météo-France surface weather stations along the coasts for atmospheric parameters), ANTILOPE to validate the rainfall accumulations and also altimetric data from satellites to complete the validation of waves parameters. On average





the simulations showed a good agreement with either atmospheric and waves observations. However, it can be noticed that both $H_s$ and $T_p$ tend to be underestimated by the model whereas the atmospheric parameters tend to be overestimated. Furthermore, the simulated convective system over sea appeared to move eastwards faster than the observed one.

Then, a sensitivity analysis was done to study the effects of waves and the impact of the atmosphere-waves coupling. It

showed large differences when the impact of the sea state was taken into account in the surface turbulent fluxes. Indeed, in AWF compared to AY, under the strong easterly wind upstream of the convective system, the generated wind-sea significantly increased the sea surface roughness length (locally up to $1 \cdot 10^{-2}$ m) and the momentum flux which resulted in a slow down of the 10m wind intensity of more than $1$ m.s$^{-1}$ over a large area. This decrease was more important than in the previous studies of Thévenot et al. (2016) and Bouin et al. (2017) due to very rough sea conditions and a strong wind regime in our studied

case. Added to that, a decrease of the latent heat flux was noticed along the French Riviera but did not represent a significant decrease of the total flux (by 2% on average). Larger differences were found under the convective system but were more likely associated to differences in its location over sea at small scale. The scores did not show significant change except for the wind intensity especially at the Azur buoy where the wind decreased in AWF and the bias was reduced.

Looking at the impact of air-wave coupling by comparing AWC to AWF, minor effects were found. Still, locally it can

be noticed that $z_0$ tends to increase (up to $2 \cdot 10^{-3}$ m) along with a slight decrease of the wind intensity by $0.6$ m.s$^{-1}$. As a feedback of the wind on waves we found a decrease of $H_s$ and $T_p$ compared to WY. For the heat fluxes, very small changes were found. Thus, the coupling effect appeared to be smaller than the forcing effect and the comparison between WY (forcing AWF) and AWC showed very close results in terms of wave modelling. However, this result must be moderated by the fact that it was a single case study and only one short-term forecast was considered. To conclude more robustly on the effect of the

coupling, additional tests with the AROME-WW3 coupled system must be conducted regarding notably the parametrizations of the WW3 configuration, the initialization or the coupling frequency.

In all simulations the rainfall amount forecast was in good agreement in the South of France over the Hérault area. It was found that in AWF the convective system over sea was shifted eastwards, during phase II, of about 40 km compared to AY. This was due to the changes in the low level dynamics in AWF in response to the sea state, that modified the position of the

convergence line.

Either on AWF or AWC the waves didn't have a significant impact on heat fluxes whereas favourable conditions were met for this with strong sea state with young sea collocated with strong turbulent heat fluxes. Thus, it seemed that the sea state cannot directly affect in a significant way the heat exchanges at the air-sea interface during Mediterranean HPEs. Nevertheless, the dynamical impact appeared relevant to take into account in high-resolution weather forecast as it affected the marine low

level flow that is a key ingredient for HPE.

Further investigations still need to be done to improve our understanding of the air-sea interactions. Future work will consist in adding the interactive evolution of the ocean and thus of the sea surface temperature (constant here during the forecast time) which is known to have an effect on the lower levels of the atmosphere. This will be done using kilometric-scale tri-coupled ocean-atmosphere-wave simulations. One objective is to quantify the impact of the ocean on the forecast compared to the





impact of waves. Finally, regarding the parametrization of the sea surface turbulent fluxes WASP, it needs to be tested and validated on more study cases, especially during strong wind conditions, in order to further assess its added value.

*Author contributions.* All authors (CS, CLB, MNB and VD) contribute to the conceptualization, methodology, writing draft preparation, review and editing of the article. MNB developed the WASP parametrization and manages its integration in the SURFEX code. Coupling
5  development and simulations was done by CS and CLB. CS, CLB and MNB proceeded to the validation and analysis of the results.

*Competing interests.* The author declare that they have no conflict of interest.

*Acknowledgements.* This work is a contribution to the HyMeX program (Hydrological cycle in the Mediterranean EXperiment - http://www.hymex.org) through INSU-MISTRALS support. The authors acknowledge the Occitanie French region for its contribution to César Sauvage's PhD at CNRM. The authors acknowledge the MISTRALS/HyMeX database teams (ESPRI/IPSL and SEDOO/OMP) for their help in accessing to
10  the surface weather station data. Oceanographic buoys data and the PSY4V3R1 daily analyses were made available by the Copernicus Marine Environment Monitoring Service (marine.copernicus.eu). Altimeter data were freely available from the CERSAT service at IFREMER. The authors gratefully acknowledge Mickaël Accensi and Fabrice Ardhuin (LOPS) for their invaluable help and advices concerning the development and application of the north-western Mediterranean configuration of the WaveWatchIII model. The authors finally thank Olivier Nuissier (CNRM) for discussions about the characteristics of the studied Mediterranean heavy precipitation event.



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



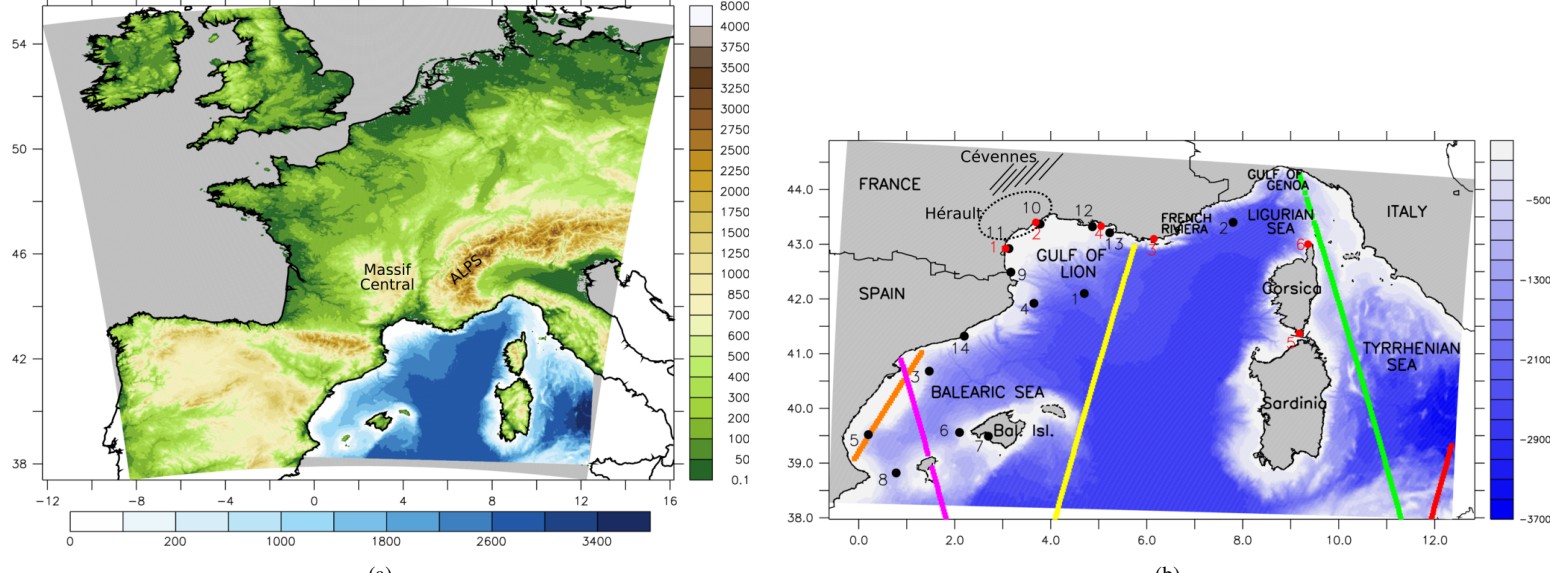

**Figure 1.** Simulation domain: (a) AROME-France (topography, m) and (b) WW3 domain illustrated with bathymetry (blue scale, m). Black circles indicate the location of the moored buoys and red circles the location of the surface stations (see Tab 1); in green the satellite track on 12/10 around 05UTC; in yellow on 13/10 around 18UTC; in magenta on 14/10 around 05UTC and in red and orange on 14/10 around 18UTC. Orange track is for JASON-2 and the others are for SARAL.

| Name | Longitude | Latitude | Source | Name | Longitude | Latitude | Source |
|---|---|---|---|---|---|---|---|
| 1-Lion | 4.7°E | 42.1°N | Météo-France | 11-Leucate | 3.13°E | 42.92°N | CMEMS |
| 2-Azur | 7.8°E | 43.4°N | Météo-France | 12-MesuRho | 4.87°E | 43.32°N | CMEMS |
| 3-Tarragona | 1.47°E | 40.68°N | CMEMS | 13-Le Planier | 5.23°E | 43.21°N | CMEMS |
| 4-Begur | 3.65°E | 41.92°N | CMEMS | 14-Barcelone | 2.2°E | 41.32°N | CMEMS |
| 5-Valence | 0.20°E | 39.52°N | CMEMS | *1-Leucate* | 3.06°E | 42.92°N | Météo-France |
| 6-Dragonera | 2.1°E | 39.56°N | CMEMS | *2-Sète* | 3.69°E | 43.4°N | Météo-France |
| 7-Bahia de Palma | 2.7°E | 39.49°N | CMEMS | *3-Hyères* | 6.15°E | 43.1°N | Météo-France |
| 8-Canal de Ibiza | 0.78°E | 38.82°N | CMEMS | *4-Martigues* | 5.05°E | 43.33°N | Météo-France |
| 9-Banyuls-sur-mer | 3.17°E | 42.49°N | CMEMS | *5-Bonifacio* | 9.18°W | 41.37°N | Météo-France |
| 10-Sète | 3.78°E | 43.37°N | CMEMS | *6-Ersa* | 9.36°W | 43°N | Météo-France |

**Table 1.** Names and locations of the moored buoys and surface stations (in italic) used for validation.



| Moored buoys and surface stations | | | | | | | | | | | | |
| | WY | | | AY | | | AWF | | | AWC | | |
| Parameter | Biais | RMSE | Correlation | Biais | RMSE | Correlation | Biais | RMSE | Correlation | Biais | RMSE | Correlation |
| $H_s$ | -0.23 | 0.53 | 0.90 | - | - | - | - | - | - | -0.28 | 0.58 | 0.90 |
| $T_p$ | -0.79 | 1.16 | 0.90 | - | - | - | - | - | - | -1.27 | 1.64 | 0.88 |
| WSP | - | - | - | 0.22 | 2.70 | 0.66 | 0.04 | 2.75 | 0.64 | 0.09 | 2.67 | 0.65 |
| WDIR | - | - | - | 1.43 | 42.05 | 0.85 | 2.0 | 42.46 | 0.86 | 1.85 | 42.95 | 0.85 |
| T2M | - | - | - | 0.39 | 1.25 | 0.70 | 0.45 | 1.32 | 0.66 | 0.44 | 1.32 | 0.66 |
| RH2M | - | - | - | 2.19 | 8.84 | 0.79 | 2.89 | 9.66 | 0.76 | 3.0 | 9.97 | 0.76 |
| Satellites | | | | | | | | | | | | |
| $H_s$ | -0.17 | 0.4 | 0.78 | - | - | - | - | - | - | -0.28 | 0.5 | 0.71 |

**Table 2.** Skill scores computed against surface stations and satellite data for the wave-only simulation (WY) and the coupled simulation (AWC) for wave parameters ($H_s$, $T_p$) and for the atmosphere-only simulations (AY, AWF) and coupled simulations (AWC) for atmosphere parameters. WSP stands for 10m-wind speed (m.s$^{-1}$), WDIR for 10m-wind direction (°), T2m for 2m-air temperature (°C) and RH2M for 2m-relative humidity (%).

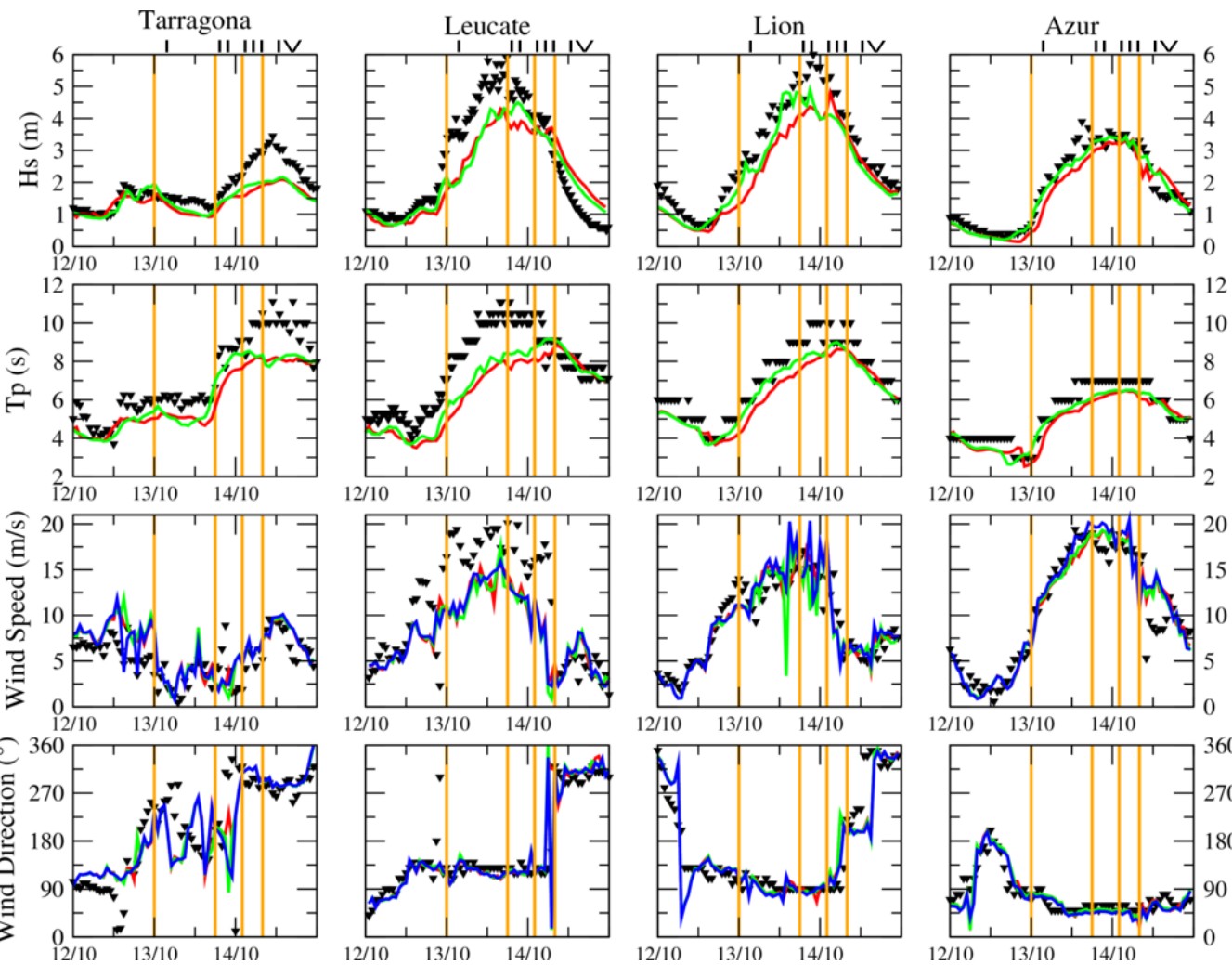

**Figure 2.** Evolution of $H_s$ (m), $T_p$ (s), wind speed (m.s$^{-1}$) and wind direction (°) simulated with AWF (green for atmospheric parameters), AWC (red), AY (blue) and WY (green for wave parameters) against buoy observations (black triangles).



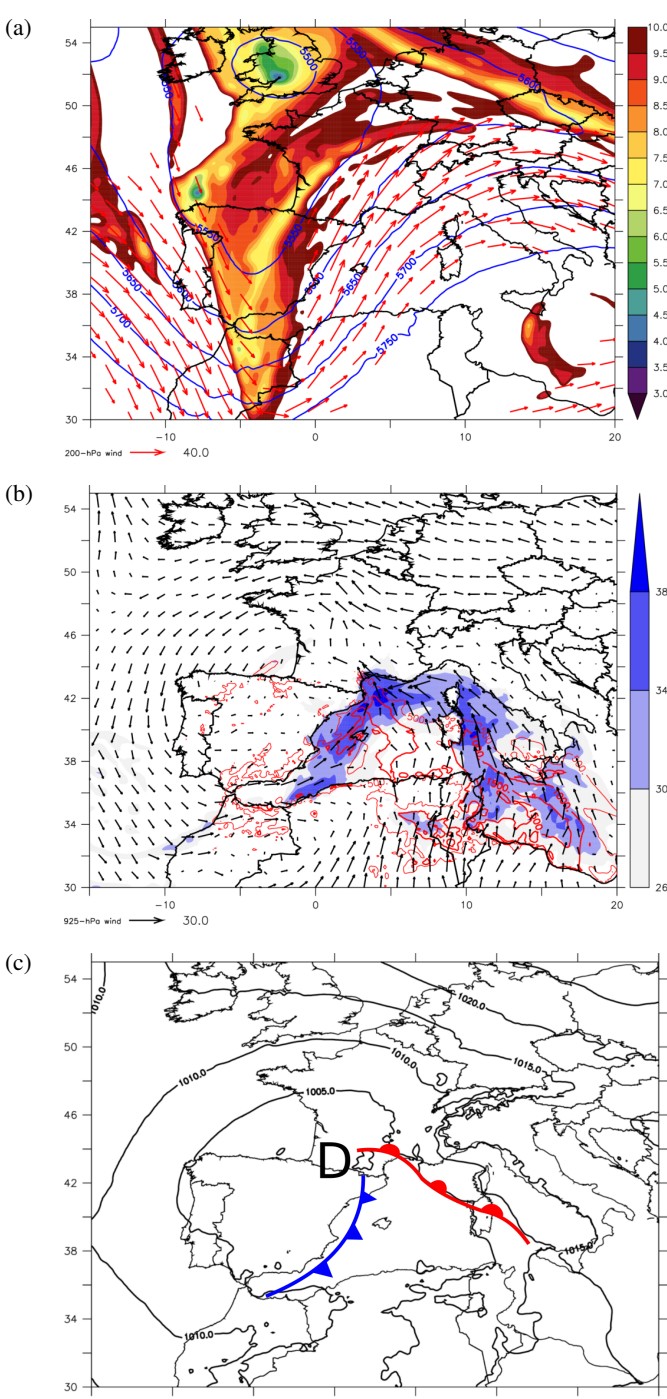

**Figure 3.** Synoptic situation at 12UTC, 13 October 2016 from the ARPEGE analysis (a) at high level: colour shading is the height of the 2 PVU iso-surface (km), blue contour is the geopotenteil height (m) at 500 hPa and black arrows the wind above 20 m/s at 200 hPa; (b) at low level: colour shading is the Integrating Water Vapor (kg.m$^{-2}$), red contour is the Convective Available Potential Energy (J.kg$^1$) and black arrows the wind at 925 hPa; (c) mean sea level pressure (black contour) and position of the cold (blue) and warm (red) fronts.





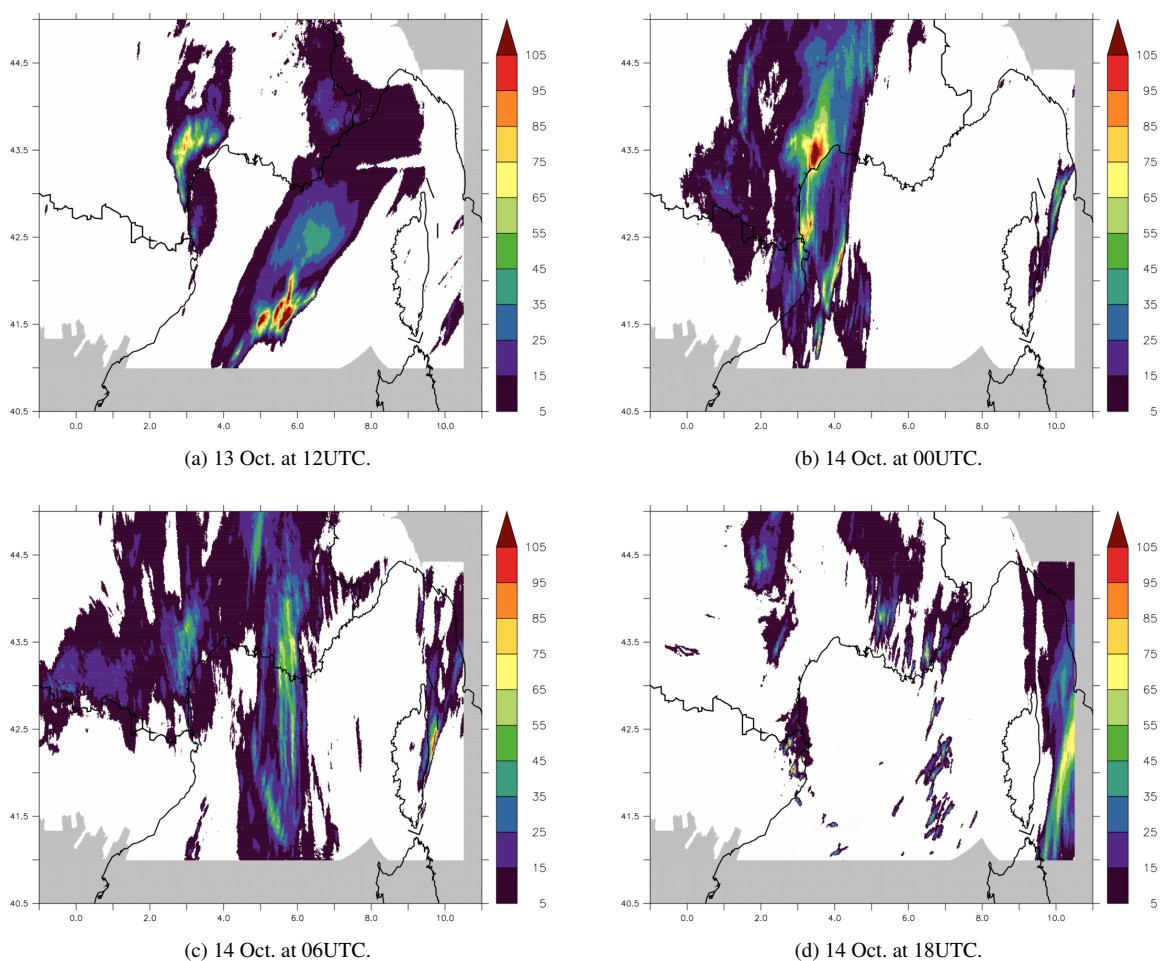

(a) 13 Oct. at 12UTC.

(b) 14 Oct. at 00UTC.

(c) 14 Oct. at 06UTC.

(d) 14 Oct. at 18UTC.

**Figure 4.** 6h-rainfall amounts (mm) from ANTILOPE observations.




**Figure 5.** (a) 6h-rainfall amount (mm); (b) Pseudo-adiabatic potential temperature $\theta_w$' (colour shading, °C) and wind (m.s$^{-1}$, arrow) at 925 hPa with the CAPE over 750 J.kg$^{-1}$ in green contour; (c) wind divergence (colour shading, $10^{-3}$ s$^{-1}$, values between -0.12 and 0.12 are masked, s$^{-1}$) at 950 hPa, black contour is the vertical velocity (Pa.s$^{-1}$) at 950 hPa and black arrows the horizontal winds (m.s$^{-1}$); (d) 10m-wind intensity and direction (m.s$^{-1}$); (e) wave age and wave direction; (f) wave significant height (m) and wave direction; (g) total turbulent heat fluxes (H, LE) (W.m$^{-2}$) and (h) wind stress (N.m$^{-2}$) simulated by AWF the 13 October at 12UTC. Blue dots in (d, e, f) stand for Tarragona, Lion and Azur buoys (west to east).





**Figure 6.** Same as Figure 5 but the 14 October 00UTC.

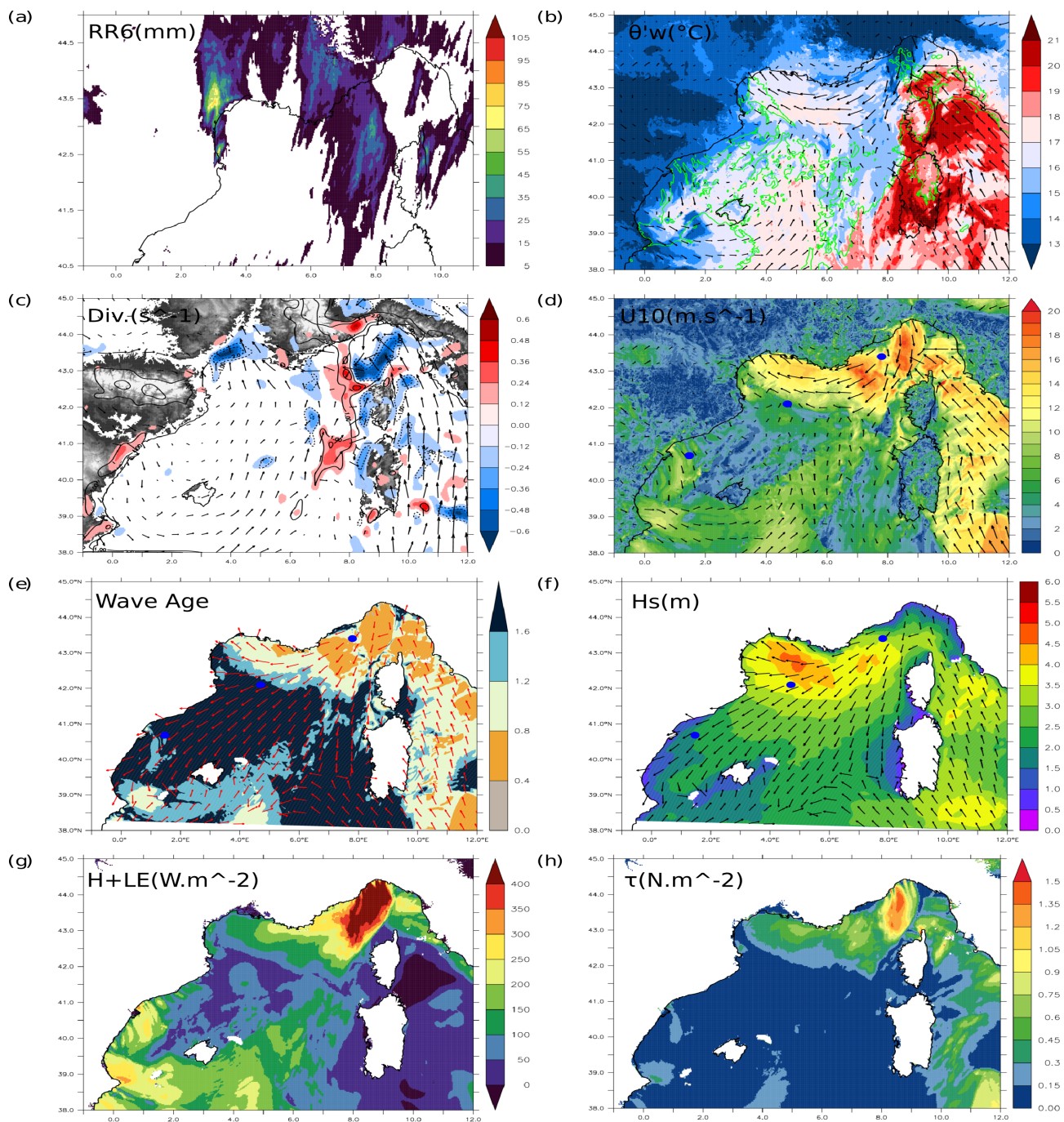

**Figure 7.** Same as Figure 5 but the 14 October 06UTC.





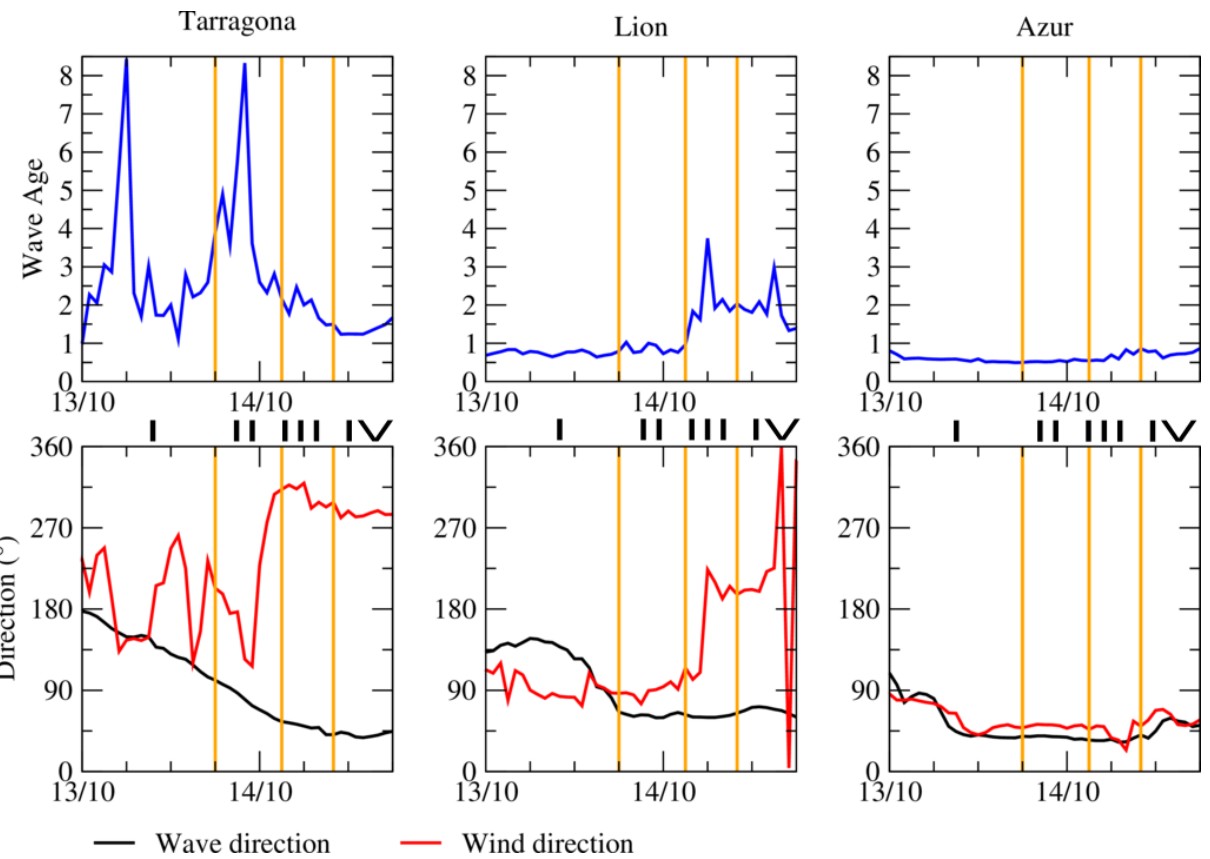

**Figure 8.** Evolution of the wage age (blue) and direction (°) of local wind (red) and waves (black) simulated with AWF (WY) during the 13 October run at three different moored buoys. Orange lines limit the four different phases (I, II, III, IV).

(a) 13 October at 14UTC $z_0$: AWF-AY

(b) 13 October at 14UTC $z_0$: AWF-AWC

(c) 14 October at 00UTC $z_0$: AWF-AY

(d) 14 October at 00UTC $z_0$: AWF-AWC

**Figure 9.** $z_0$ ($10^{-3}$ m) differences for the 13 October at 14UTC (a,b) and the 14 October at 00UTC (c,d) between AWF-AY (a,c) and AWF-AWC (b,d).




(a) 13 October at 14UTC wind: AWF-AY

(b) 13 October at 14UTC wind: AWF-AWC

(c) 14 October at 00UTC wind: AWF-AY

(d) 14 October at 00UTC wind: AWF-AWC

**Figure 10.** Wind (m.s$^{-1}$) differences for the 13 October at 14UTC (a,b) and the 14 October at 00UTC (c,d) between AWF-AY (a,c) and AWF-AWC (b,d).







**Figure 11.** (a,b) LE (W.m$^{-2}$) and (c,d) H (W.m$^{-2}$) differences for the 14 October at 00UTC between AWF-AY (a,c) and AWF-AWC (b,d).



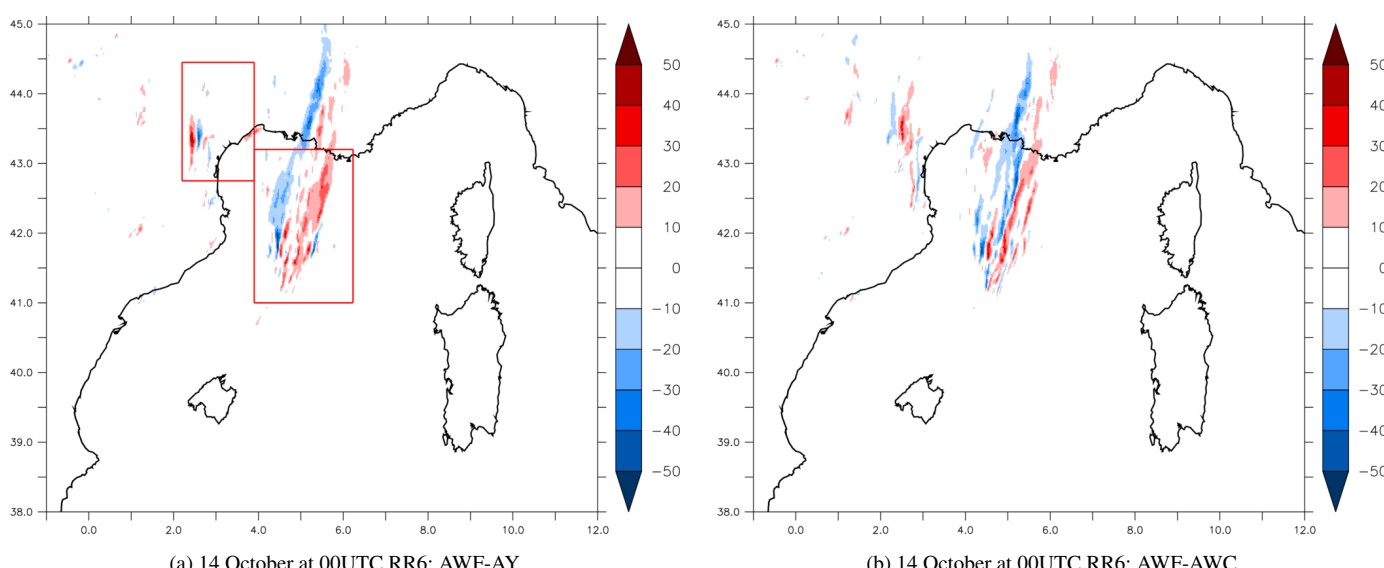

(a) 14 October at 00UTC RR6: AWF-AY     (b) 14 October at 00UTC RR6: AWF-AWC

**Figure 12.** 6h-rainfall amount (mm) differences for the 14 October at 00UTC between AWF-AY (a) and AWF-AWC (b).