# Peer review of "Characterization of the air-sea exchanges mechanisms during a Mediterranean heavy precipitation event using realistic sea state modelling"

_Atmospheric Chemistry and Physics, 2019_

## Referee Comment (RC1) · Anonymous Referee #1 · 30 Sep 2019

Review of: Characterization of the air-sea exchanges mechanisms during a Mediterranean heavy precipitation event using realistic sea state modelling César Sauvage 1 , Cindy Lebeaupin Brossier 1 , Marie-Noëlle Bouin 1,2, and Véronique Ducrocq 1

General comments

=============

This paper describes a case study assessment of the introduction of an interactive wave simulation to describe sea state as a lower boundary to an atmosphere model

simulation of a heavy rainfall event over the Mediterranean. The Introduction and Model Description sections are in general very clear and efficiently set out. Results are compared with a comprehensive set of surface and satellite-based observations of atmosphere and wave variables. As set out below, aspects of the experimental design and dependence on a single case limit the extent to which the paper can add value to the existing literature, and the authors are encouraged to consider this further. Overall, the paper is generally written to a good standard, is relevant and has scientific merit.

I am content that the paper should be published following some minor, but significant(!), suggested corrections and considerations as set out below. Each in themselves are perhaps worthy of 'major' corrections and further work, as acknowledged by the authors in the concluding paragraph, but perhaps it is sufficient that the choices made are more directly addressed and justified within the current paper rather than recommending a more significant re-write and further simulation and analysis work.

Specific comments

===============

1. Wave model coupling approach
* * *
Section 2.3.2 – wave model coupling. The authors describe the use of the WASP parameterisation of the surface roughness and coupling via the wave model simulated peak period. This seems like a rather indirect approach, given that more typically the WAVEWATCHIII calculated Charnock parameter could be used directly into Eq. 7 (e.g. Varlas et al., 2018; Section 2). In fact, Wahle et al. 2017 pass the WAM wave model calculated roughness length directly (see their Section 2.3). The direct use of WAVEWATCHIII computed Charnock parameter was also described for example in coupling studies of the North West European shelf by Lewis et al (2018, 2019).

Another study cited, Renault et al., 2012, apply a similar wave-age dependent coupling

(their Section 3.5), and here reference to Drennen et al., 2005 might be appropriate. Further, more detail of p5,l30 ("coefficients A and B being polynomial functions of the surface wind speed") would be useful.

The authors should set out their rationale for the WASP parameterisation in preference to the wave model computed Charnock or roughness. Indeed, a comparison between the WASP and Wavewatch computed roughness would have been a very enlightening addition to this discussion and of wider use for assessing potential modelling uncertainties for the community. In short, what is the sensitivity of results (roughness lengths) to this configuration choice?

2. A-W coupling experiments

————————————————

Section 2.4 – set of experiments. The authors set out the 4 (WY, AY, AWF and AWC) experiments. On the one hand, this is a justifiable and clean experimental design. However, given the increasing use of more fully coupled atmosphere-ocean-wave regional configurations for similar case study assessments (e.g. Renault et al, Ricci et al, etc), the authors should more directly justify the lack of ocean interactions within the current study. This is highlighted in the final paragraph of the paper, but should also be addressed directly in the choice of experiments described in Section 2.

Finally, please comment on expected sensitivity of results to the choice of coupling frequency (1h). Were any sensitivity tests conducted to assess this? Some studies (e.g. Renault et al., though many others exist), involve interactions at much higher coupling frequency, to capture interactions with fast moving systems for example.

3. Simulation lead time considerations

——————————————————-

The authors chose to validate only the first 24h of each ARMOME simulation in Section 3, though simulations covered T+0 to T+42. Why are data beyond the first day not

considered? Similarly, the focus in Section 5 is on T+14 and T+24 snapshots only.

Converse to this, would you expect the impact of wave interactions to perhaps grow with time (some spin up effect) if all regional simulations were initialised from the same operational analysis? Please also comment on the time taken for 1.3 km scale high-resolution details to spin up within the model domain. This spin up effect may help explain the rather similar results shown in Fig. 2.

Wave results for AWF seem slightly degraded relative to WY in Fig 2., but not commented on. Is there some explanation for the different behaviour?

In Section 5, what is the sensitivity to model lead time? Presumably there are periods of overlapping data from different model start times for these periods of interest? Does the influence of wave interaction grow with lead time, or are results dominated by increasing errors? Authors state that "differences between three simulations were well established", but are you confident differences were spun up?

In general, all simulation results seem to be essentially similar, and it is difficult to assess how much this is a true reflection that the systems are not very sensitive to wave interactions (a null result, which should be more explicitly captured in the Abstract), or a symptom of the experimental design. Authors should be clearer in their discussion on this.

4. Discussion of precipitation differences

–––––––––––––––––––––––––––––––-

The overall conclusion from Section 5.2 appears to be that simulations were "about the same". It is again difficult to judge the extent to which differences just reflected expected variability in the simulation (e.g. as might be reflected in an ensemble of simulations of the case), and how much any differences could be attributed more physically to changes in low-level flows and heat fluxes previously described. P13, l33 should therefore be expanded to provide a more qualified discussion of how "this displacement was directly linked to….". I am otherwise left with an impression that the precipitation differences are somehow 'random' and could equally be produced with some other change in (e.g.) model parameters, initial condition etc. There is some attempt at this in the summary from l5, p14, but this could be more explicitly set out. For example, phrases like "due to differences in terms of heat fluxes…" is too vague here to help the reader follow the physical arguments being discussed.

It might be instructive to discuss the relative sensitivity of the system, e.g. with reference to any operational ensemble information available at the time of this particular study, to set some context.

5. Dependence on a single case study

————————————————————-

It is difficult to assess the significance of this paper to a general readership and to the community, given that it addresses only a single case study. However, it is equally not clear how many such cases would need to be considered before some robust statistics are achieved, and the key physical mechanisms are lost in the number of cases addressed – it would be a quite different paper in fact.

The authors should however be clearer, perhaps in both the methods and discussion sections, on the relevance of the single case to wider improvement of understanding and simulation quality for the region. What can operational centres learn (if anything) from the study for development of forecast model configurations for example? Suggesting that further cases are considered in a similar manner would fundamentally change the submitted manuscript, so is not recommended by this reviewer, but the limitations of the single study (particularly assessed in a deterministic framework) should be more openly acknowledged and discussed. Further, discussion of how the current paper adds value beyond some earlier work in the region (e.g. Renault et al, 2012 and later references) would be welcome.

Technical corrections

==================

P6,l23 – do you mean $\frac{1}{2}$ deg. or perhaps 1/12 deg. global WW3 resolution model? Could not work out if the boundary conditions were rather coarse scale (and if so, please comment on any boundary spin up issues into much higher resolution system), or if a typo.

P6, l27 – please comment if SST is updated daily (as implied) during the simulation, or a fixed SST is used throughout the 42h simulation? One might again expect precipitation fields to be rather sensitive to details (e.g. resolution, updating frequency) of the SST field in this case (e.g. Lebaupin Brossier et al, 2006; 2008) – authors should comment. Useful to also confirm if any surface currents information is used, or if assumed to have a stationary sea surface?

Fig. 9, 10, 11, 12 – would be clearer to plot impact of interactive coupling differences as (AWC – AWF) in panels b) and d), given panels a) and c) establish differences of AWF to AY. The additional impact of coupling here is the inverse of what is currently shown, so is a bit confusing to follow. For example, in Fig. 12, are the AWC differences just the inverse of AWF to AY (such that AWC is more similar to AY than AWF?), or is the main rain area further displaced again?
* * *

---

## Referee Comment (RC2) · Anonymous Referee #2 · 13 Oct 2019

This paper describes how to directly add the impact of ocean waves in the parameterisation of the aerodynamical roughness length scale used by an atmospheric model to prescribe the momentum exchange between the atmosphere and the sea surface. A single case study is then used to illustrate the impact on short range prediction of a heavy precipitation event. This type of work is not entirely new as I would encourage the author to refer to Peter Janssen book (Janssen 2004), which clearly supports the concept of two-way active coupling between an operational atmospheric model and a wave model. What is novel and worthy of publication is the WASP parameterisation.

[Figure]

For this reason, the actual expression of A and B would be a very good addition to the paper. However, the authors need to explain a bit more their choice of the WASP parameterisation, rather than using the Charnock values that WW3 can produce. A comparison of the WW3 Charnock and the WASP counterpart will be required and any major differences would need to be justified. Anemometers mounted on buoys are rarely at 10m height. Nothing is mentioned regarding the adjustment of the buoy winds to 10m. The discussion regarding the bias reduction of 10m winds is only relevant if the buoy winds have been adjusted to 10m. Revise the manuscript accordingly. The peak wave period is not a very stable quantity in situation of multi peak wave spectra. In the Mediterranean Sea, it is not too often the case, but over open ocean conditions, this is more often the norm. How would the WASP parameterisation deal with such situation? Is Tp computed from the full 2D-spectrum? Should one instead only determine Tp from the windsea part of the spectrum? Obviously, this paper is only a one case study. It has focussed on short range forecasts. This needs to be clearly highlighted and discussed. In Janssen (2004), the impact of the coupling to waves is shown to be even more important at longer lead time. In the final section, it is discussed that ocean waves have an impact on the momentum flux across the air-sea interface. However, according to Janssen and Bidlot (2018), waves might also impact on the latent and sensible heat fluxes https://doi.org/10.1016/j.piutam.2018.03.003 https://www.sciencedirect.com/science/article/pii/S2210983818300038

Minor corrections: At a few places: biais -> bias P2, line 26: waves, known as sea spray, occurs -> waves occurs, generating sea spray P3, line 2: add Janssen 2004 P4, line 9: the adjustments of Bidlot et al. are only relevant if you use ST3, otherwise with ST4, there is an all new prescription of the whitecap dissipation that does not use Bidlot et al. P5, in (7), the first term is the Charnock relation, the second term is the viscous contribution to z0. See Beljaars, A. C. M. (1994). The parametrization of surface uxes in large-scale models under free convection. Q. J. R. Meteorol. Soc., 121, 255{270.

P5, line 28: the Charnock parameter is -> the surface roughness is P6, line 15: to an

untrained person, the relation between Tp and Ua would appear to be not correct. Truly speaking, one can find a relation between Tp $\sim$ c * Ua/g, with c non dimensional and from empirical fetch relation find that for a typical non dimensional fetch c $\sim$ 5, hence why one can simply write Tp $\sim$ 0.5 Tp P8, line 7: well represents -> represents well P8, line 8: It also can be noticed a delay -> Also, there is a delay P14, line 18: In this purpose -> For this purpose P14, line 21: coupled way -> coupled mode P14, line 30: affected all the event – affected during all the event
* * *

---

## Author Comment (AC1) · 10 Dec 2019

**REFEREE#1**

This paper describes a case study assessment of the introduction of an interactive wave simulation to describe sea state as a lower boundary to an atmosphere model simulation of a heavy rainfall event over the Mediterranean. The Introduction and Model Description sections are in general very clear and efficiently set out. Results are compared with a comprehensive set of surface and satellite-based observations of atmosphere and wave variables. As set out below, aspects of the experimental design and dependence on a single case limit the extent to which the paper can add value to the existing literature, and the authors are encouraged to consider this further. Overall, the paper is generally written to a good standard, is relevant and has scientific merit.

I am content that the paper should be published following some minor, but significant(!), suggested corrections and considerations as set out below. Each in themselves are perhaps worthy of 'major' corrections and further work, as acknowledged by the authors in the concluding paragraph, but perhaps it is sufficient that the choices made are more directly addressed and justified within the current paper rather than recommending a more significant re-write and further simulation and analysis work.

**Specific comments**

1. Wave model coupling approach
   ───────────────────────────────

→ Section 2.3.2 – wave model coupling. The authors describe the use of the WASP parameterisation of the surface roughness and coupling via the wave model simulated peak period. This seems like a rather indirect approach, given that more typically the WAVEWATCHIII calculated Charnock parameter could be used directly into Eq. 7 (e.g. Varlas et al., 2018; Section 2). In fact, Wahle et al. 2017 pass the WAM wave model calculated roughness length directly (see their Section 2.3). The direct use of WAVEWATCHIII computed Charnock parameter was also described for example in coupling studies of the North West European shelf by Lewis et al (2018, 2019).
Another study cited, Renault et al., 2012, apply a similar wave-age dependent coupling (their Section 3.5), and here reference to Drennen et al., 2005 might be appropriate.

→ Further, more detail of p5,l30 ("coefficients A and B being polynomial functions of the surface wind speed") would be useful.

→ The authors should set out their rationale for the WASP parameterisation in preference to the wave model computed Charnock or roughness. Indeed, a comparison between the WASP and Wavewatch computed roughness would have been a very enlightening addition to this discussion and of wider use for assessing potential modelling uncertainties for the community. In short, what is the sensitivity of results (roughness lengths) to this configuration choice?

Both reviewers suggest that a more direct wind-wave coupling approach would be to make use of the Charnock parameter which is computed in the wave models. This is indeed the classical approach for atmosphere-wave coupling.
The computation of the Charnock parameter within wave models is in fact known to be very sensitive, through the wind input ($S_{in}$), to the high-frequency tail of the spectrum (see Eq. 5.22 and 5.24 in Janssen, 2004), which is always parameterized in wave models, because high frequencies cannot be represented explicitly (in $f^5$ in WW3 and WAM). Some sensitivity tests on WW3 and WAM also showed that there is a small variability in the Charnock parameter due to the wave field variability at a given wind speed. Thus, the benefit of coupling with a wave model is reduced. This point is notably discussed in Voldoire et al. (2017).

The WASP approach used here has two advantages, compared to the Charnock parameter approach: i) it allows to compare more directly the wave parameter coupled fields with observations, and so to check their validity; ii) the Charnock parameter is defined differently depending on the wind speed range considered, enabling to represent in a more physical way its behaviour and possible dependency on the waves. Especially, it reproduces the observed decrease of the drag coefficient by very strong wind, which would not be possible using a wave-age only Charnock parameter as in Drennan et al. (2005). The WASP parametrization, unlike those based on wave-age Charnock parameters, is then usable for wind speeds up to 50 m s$^{-1}$.

In more detail, the Charnock parameter is calculated in WASP as follows :
- For wind speed at first level (Ua) below 7 m/s, it is a power function of Ua: $\alpha ch = aUa^b$, with a=0.7 and b=-2.52;
- For Ua above 7 m/s, the dependency to wave-age ($\chi$) is introduced and defined as: $\alpha ch = A\chi^B$, where A and B are polynomial functions of Ua:
$$A=A_0+A_1Ua+A_2Ua^2+A_3Ua^3$$
$$B=B_0+B_1Ua+B_2Ua^2+B_3Ua^3$$
detailed in Table A.

| | A0 / B0 | A1 / B1 | A2 / B2 | A3 / B3 |
|---|---|---|---|---|
| $7 \leq Ua < 23$ | -9.202 / -4.124E-1 | 2.265 / -2.225E-1 | -1.340E-1 / 1.178E-2 | 2.350E-3 / 1.616E-4 |
| $23 \leq Ua < 25$ | 2.270 / -2.410 | 6.670E-2 / 4.300E-2 | 0.0 / 0.0 | 0.0 / 0.0 |
| $Ua \geq 25$ | 9.810E-2 / 0.0 | -4.130E-3 / 0.0 | 4.340E-5 / 0.0 | 1.160E-8 / 0.0 |

Table A: Coefficients of the polynomial functions A and B, depending on the wind speed range.

Figure A shows the comparison between the Charnock parameter calculated with WASP and with WW3 the 14 October at 00UTC. It highlights that, by using WASP, we obtain much higher variability in the Charnock parameter.

Section 2.3.2 has been enlarged to better describe these advantages of using the WASP parameterization and an appendix has been added to the paper with the details of the calculation of the Charnock parameter (as shown above).

[Figure]

*Figure A: Charnock parameter from (a) WASP and (b) WW3 the 14 october at 00UTC.*

[Figure]

*Figure B: Charnock coefficient from (a) WASP as a function of 10 m-wind (m/s) and from (b) WW3*
*for 13 october 2016 (grey dots and mean value (+/- one standard deviation) in red).*

**2. A-W coupling experiments**

Section 2.4 – set of experiments. The authors set out the 4 (WY, AY, AWF and AWC) experiments. On the one hand, this is a justifiable and clean experimental design.

→ However, given the increasing use of more fully coupled atmosphere-ocean-wave regional configurations for similar case study assessments (e.g. Renault et al, Ricci et al, etc), the authors should more directly justify the lack of ocean interactions within the current study. This is highlighted in the final paragraph of the paper, but should also be addressed directly in the choice of experiments described in Section 2.

This is a good remark from the reviewer. Indeed, there is no ocean coupling in the experiments presented in the paper.
The objective of this study is to better assess the role of the waves on the dynamics (i.e. the impact

on the momentum flux and surface wind) but also the impact on the sea surface turbulent heat fluxes during this kind of Mediterranean heavy precipitation event. The ocean interactive coupling by modifying the SST during the model integration has a significant impact on the heat and moisture fluxes (e.g. for Mediterranean HPE Lebeaupin Brossier et al. 2009 and Rainaud et al. 2017) that could mask the effect of the waves on these fluxes. So, a fixed SST is used here. Also, this way, the AY experiment is also very close to the AROME operational forecast which uses a fixed SST field.
A comment has been added at the beginning of section 2.4.

We would like to mention here that a tri-coupled experiment AROME-NEMO-WW3 of the same case is currently under investigation in order to analyse the different coupled impacts.

→ Finally, please comment on expected sensitivity of results to the choice of coupling frequency (1h). Were any sensitivity tests conducted to assess this? Some studies (e.g. Renault et al., though many others exist), involve interactions at much higher coupling frequency, to capture interactions with fast moving systems for example.

The coupling frequency was set to 1h to be consistent with our forced experiment (AWF) as the AROME forcing is routinely available hourly and thus to compare fairly AWC and AWF. We didn't test any higher or lower coupling frequency.
Also, the variability of wave parameter and wind appears quite well captured with a frequency of 1h, as shown for example in Figure 2 of the paper.
Higher coupling frequency would possibly have introduced more variability in the results and we agree this should be carefully tested in the future, along with the use of instantaneous or averaged wind/wave coupled fields.

3. Simulation lead time considerations
——————————————————————-
→ The authors chose to validate only the first 24h of each AROME simulation in Section 3, though simulations covered T+0 to T+42. Why are data beyond the first day not considered? Similarly, the focus in Section 5 is on T+14 and T+24 snapshots only.

Indeed, for each day, for the period from 00UT to 18UT we have two forecasts available, except for the wave-only simulation WY which is a continuous simulation. So to fairly validate the wave parameters in WY/AWF and AWC, we only considered the first 24 hours of each forecast, and to be consistent in the atmospheric component validation, we examined the same forecast range ( T+0 to T+24h) for wind in AY, AWF, AWC.
In section 4, we examine the forecast starting on 13 october 2016 00UT, in order to have the continuity necessary to well describe the event, in particular the initiation mechanisms and the mature/strongly precipitating phase.
In section 5, we only show the range T+14 to T+24 as this section aims to focus on the Phases I (initiation) and II (mature phase) of the event, when the atmospheric conditions and sea state appear more favorable to strong air-wave interactions, and so when coupling may more strongly impact the convective system triggering and sustaining. But, in fact the whole forecast starting on 13 october 2016 00UT is used and the Phases III and IV were also examined, but they are more representative of the system moving eastward and intensity decreasing over France.

→ Converse to this, would you expect the impact of wave interactions to perhaps grow with time (some spin up effect) if all regional simulations were initialised from the same operational analysis? Please also comment on the time taken for 1.3 km scale high-resolution details to spin up within the model domain. This spin up effect may help explain the rather similar results shown in Fig. 2.

From the atmospheric point of view, all the experiments start from the AROME analysis at 00UTC which is also at a 1.3 km resolution. The spin-up is so quite short (<1h).
Only the wave conditions at the surface vary between the experiments.
But, AWF and AWC started from the same sea state the 12 october at 00UTC. This state is extracted from WY after 7 days of simulation (between 5 and 12 october) that was enough to spin up the high-resolution wave model.
In conclusion, the impact of wave interactions may grow with time during the forecast and also progressively from one forecast to another as WaveWatchIII restarts each time at 00UTC from the previous forecast at T+24h. But the experimental design used here, with the same atmospheric initiation in AWC and AWF, can not fully clarify this.
To properly consider this issue, it would be necessary to run longer experiments (more than 3 successive forecasts) and also to include the production of new atmospheric analysis using the previous forecast (of AWF or AWC) as background.
This point is now included in the conclusion.

→ Wave results for AWF seem slightly degraded relative to WY in Fig 2., but not commented on. Is there some explanation for the different behaviour?

Actually, waves in AWF and WY are exactly the same.
Concerning the differences between AWC and WY, there is indeed a small degradation of the results with decreases of Hs and Tp with coupling. The studies of Renault et al. 2012 and of Whale et al. 2017 showed also, negative feedbacks on Hs and Tp, associated with reduced wind when coupled.
The caption of Figure 2 has been changed to avoid confusion between simulations.

→ In Section 5, what is the sensitivity to model lead time? Presumably there are periods of overlapping data from different model start times for these periods of interest? Does the influence of wave interaction grow with lead time, or are results dominated by increasing errors? Authors state that "differences between three simulations were well established", but are you confident differences were spun up?

As previously said, the impact of wave interactions may grow with time during the forecast and also progressively from one forecast to another as WaveWatchIII restarts each time from the previous forecast. Figures C and D show the evolution of the differences of Tp and surface wind between the coupled experiment AWC and the forced one AWF/WY. As already explained, the AROME analysis is used as initial conditions for both AWF and AWC. In fact for the overlapping periods there is no clear evidence of error due to the lead time considered. The differences in terms of wind are maximum and large when convective systems affect the considered area, but do not clearly grow with time. In terms of Tp, the differences between AWC and AWF/WY are maximum but limited to -1.4s the 13 october, then tend to zero at the end of 14 october.
In Section 5, we analyse the 42 hours of forecast starting from the 13 October at 00UTC, in order to examine in a continuous way the mechanisms at the air-sea interface and the effects of waves on the HPE. Also, the mature phase of convective systems was identified from 19UTC, 13 october to 03UTC, 14 october, thus without overlapping most of the time.
The introduction of section 5 has been rewritten to clarify this.

[Figure]

*Figure C: Tp (s) differences evolution between AWC and AWF(WY) from 12 to 15 october 2016 00UTC at the Tarragona, Leucate, Lion and Azur buoys: The forecast starting on 12 october 2016 00UTC is in black, the one starting on 13 in red and the one starting on 14 in blue.*

[Figure]

*Figure D: Same as Figure C but for the 10m-wind speed (m/s) between AWC and AWF.*

→ In general, all simulation results seem to be essentially similar, and it is difficult to assess how much this is a true reflection that the systems are not very sensitive to wave interactions (a null result, which should be more explicitly captured in the Abstract), or a symptom of the experimental design. Authors should be clearer in their discussion on this.

In fact the system is very sensitive to waves especially on the dynamics, this might need to be made clearer in the paper. The slow-down of the 10m-wind along the French Riviera where the wind sea is created is over a very large area and is directly linked to the wind-wave interactions. This phenomenon was found in previous studies (i.e. Renault et al. 2012, Whale et al. 2017, Bouin et al. 2017) and so was expected here as for the displacement of the precipitation which could be also expected (Bouin et al. 2017). However, the effect of the waves on heat fluxes were not clearly assessed in previous studies. We found that indeed the heat fluxes during those mediterranean events are not very sensitive to sea state even if the studied case appeared initially suitable to expect some effect due to waves (i.e. a wide zone of young waves, large wind speed and strong sea surface fluxes below the easterly flow). With this we are now able to conclude that the dynamics of the system is really sensitive to sea state conditions but very likely that the sea surface heat fluxes are not.
Possibly, the impression that the simulation are very similar comes from our analysis of the coupling impact that indeed does not modify the atmospheric forecast so largely when compared to the forecast with wave forcing, as coupling progressively balances the wind sea, the stress and the

near-surface wind. We try to clarify this in the conclusion/discussion and the abstract.

4. Discussion of precipitation differences
————————————————————————-

The overall conclusion from Section 5.2 appears to be that simulations were "about the same". It is again difficult to judge the extent to which differences just reflected expected variability in the simulation (e.g. as might be reflected in an ensemble of simulations of the case), and how much any differences could be attributed more physically to changes in low-level flows and heat fluxes previously described. P13, l33 should therefore be expanded to provide a more qualified discussion of how "this displacement was directly linked to. . ..". I am otherwise left with an impression that the precipitation differences are somehow 'random' and could equally be produced with some other change in (e.g.) model parameters, initial condition etc. There is some attempt at this in the summary from l5, p14, but this could be more explicitly set out. For example, phrases like "due to differences in terms of heat fluxes. . ." is too vague here to help the reader follow the physical arguments being discussed. It might be instructive to discuss the relative sensitivity of the system, e.g. with reference to any operational ensemble information available at the time of this particular study, to set some context.

The operational AROME ensemble available in October 2016 was designed with AROME at 2.5 km horizontal resolution, with different initial time (03, 09, 15 and 21UTC), with the unperturbed AROME SST analysis and the operational ECUME sea surface fluxes parametrization. Thus comparing our simulations to the ensemble forecasts appears to be very complex.

In fact, we made several sensitivity experiments with AROME changing the atmospheric initial conditions (ARPEGE analysis), the sea surface fluxes parametrizations (the operational one, ECUME, its new version ECUME6 and COARE 3.0) and the initial SST fields (ARPEGE analysis, AROME analysis) partly presented in the communication from Sauvage et al. 2018 (available here: https://ams.confex.com/ams/23BLT21ASI/mediafile/Manuscript/Paper345111/ Extended_Abstract_AMS.pdf).
All these simulations result in the formation of two convective systems: one over the Hérault area and one over sea. For this latter, the large modifications of the heat fluxes induced by SST or parametrization changes are of primary importance, as they affect both the intensity and the location of the system.
In the sensitivity experiments presented in the paper, the heat fluxes are in fact not significantly modified. This result was not really expected, but it finally permits to be confident in the impact of waves which appears to be limited to a dynamical effect that displaces the convergence and not related indirectly to large modification of the convective system intensity.
The conclusion has been revised to better explain the results.

5. Dependence on a single case study
————————————————————————-

It is difficult to assess the significance of this paper to a general readership and to the community, given that it addresses only a single case study. However, it is equally not clear how many such cases would need to be considered before some robust statistics are achieved, and the key physical mechanisms are lost in the number of cases addressed – it would be a quite different paper in fact.
The authors should however be clearer, perhaps in both the methods and discussion sections, on the relevance of the single case to wider improvement of understanding and simulation quality for the region. What can operational centres learn (if anything) from the study for development of forecast model configurations for example? Suggesting that further cases are considered in a similar manner would fundamentally change the submitted manuscript, so is not recommended by this reviewer, but the limitations of the single study (particularly assessed in a deterministic framework) should be

more openly acknowledged and discussed. Further, discussion of how the current paper adds value beyond some earlier work in the region (e.g. Renault et al, 2012 and later references) would be welcome.

This is a good remark from the reviewer and we now discuss more the fact that the results are valid for our studied case in a deterministic framework and that other HPEs (and even other kinds of meteorological situations) are needed to further conclude on the sea state impact on high-resolution weather forecast.

Nevertheless, this study marks a new step in our understanding of the sea state impact on Mediterranean HPE and in the evaluation of the importance of the wind-wave interactions for convection-permitting NWP models, after the studies of Thevenot et al. 2016 and Bouin et al. 2017 that highlighted:
- a slowdown of the low-level wind, due to higher surface roughness increasing the momentum flux (even in a moderate-wind context);
- differences in the low-level dynamics influences the positioning of the convergence directly or indirectly as it modifies the propagation of cold pools over sea, and consequently the location of the heaviest precipitation.

The main results of these two previous studies are thus confirmed in using another HPE, the case of october 2016, that was particularly interesting due to the very strong wind regime at low-level generating a wind sea, favorable to large air-wave exchanges. Also, here we analyse more deeply the impact on the heat and moisture fluxes, and use an interactive atmosphere-wave coupling at a kilometer scale.

This point is now included in the conclusion/discussion section.

Technical corrections
=================
→ P6,l23 – do you mean 1/2 deg. or perhaps 1/12 deg. global WW3 resolution model? Could not work out if the boundary conditions were rather coarse scale (and if so, please comment on any boundary spin up issues into much higher resolution system), or if a typo.

Yes, the WW3 boundaries come from a 1/2° resolution WW3 global model.
In fact the wave boundary conditions are applied as classically for regional WW3 model, *i.e.*:
- boundary data are composed of 8 spectral points (each defined following 24 directions and 31 frequencies) chosen the closest as possible to our borders among the outputs points available from the 1/2° resolution WW3 global model. The coordinates of these points are the following : 0.5°E-37.5°N ; 2.5°E-37.5°N ; 5.5°E-39.5°N ; 11°E-38°N ;11°E-40°N ;11°E-40.5°N ;12.5°E-39.5°N ;13.5°E-39.5°N
- The 8 spectral points are then linearly interpolated on the regional grid (at 1/72°-resolution) in WW3 pre-process routine, and further used in the forecast run.

→ P6, l27 – please comment if SST is updated daily (as implied) during the simulation, or a fixed SST is used throughout the 42h simulation? One might again expect precipitation fields to be rather sensitive to details (e.g. resolution, updating frequency) of the SST field in this case (e.g. Lebaupin Brossier et al, 2006; 2008) – authors should comment. Useful to also confirm if any surface currents information is used, or if assumed to have a stationary sea surface?

As explained before, we used a fixed SST during each 42h-forecast. No surface current and no sea level information is taken into account.

The SST fields come from the global daily analysis of Mercator Océan International (1/12°-resolution PSY4 system), i.e. the analysis of 12 october 2016 for the forecast starting on 12 october 2016 at 00UTC, of 13 for the one starting on 13, 00UTC, etc. These SST fields present quite fine structures (Fig. E), signatures of the complex circulation in the north-western Mediterranean Sea, and are updated in the PSY4 system considering atmospheric forcing and thanks to data assimilation.

Of course, the SST controls largely the heat and moisture fluxes and thus has a strong influence on the precipitating system, as highlighted by numerous studies in the literature (e.g. Pastor et al., 2001; Lebeaupin et al., 2006; Miglietta et al., 2011; Romero et al., 2015; Stocchi and Davolio, 2016; Rainaud et al. 2017; Meroni et al. 2018a,b ; Strajnar et al. 2019; Senatore et al. 2019). And this is why in order to isolate the wave impact in this study, the SST is the same in the sensitivity experiments and no ocean coupling is applied.

We try to make this point clearer in the section 2.4.

[Figure]

*Figure E: SST field (°C) from the PSY4 analysis for 13 October 2016.*

→ Fig. 9, 10, 11, 12 – would be clearer to plot impact of interactive coupling differences as (AWC – AWF) in panels b) and d), given panels a) and c) establish differences of AWF to AY. The additional impact of coupling here is the inverse of what is currently shown, so is a bit confusing to follow. For example, in Fig. 12, are the AWC differences just the inverse of AWF to AY (such that AWC is more similar to AY than AWF?), or is the main rain area further displaced again?

Figures 9 to 12 have been changed to show the differences AWC-AWF.
Actually, in Fig. 12 the rain patch in AWC is displaced (4 km) slightly to the west compared to AWF, making it closer to the one simulated in AY (see Figure F below).

[Figure]

*Figure F: 6h-rainfall amount (mm) differences the 14 October 2016 at 00UTC between AWC-AY.*

**REFEREE#2**

This paper describes how to directly add the impact of ocean waves in the parameterisation of the aerodynamical roughness length scale used by an atmospheric model to prescribe the momentum exchange between the atmosphere and the sea surface. A single case study is then used to illustrate the impact on short range prediction of a heavy precipitation event. This type of work is not entirely new as I would encourage the author to refer to Peter Janssen book (Janssen 2004), which clearly supports the concept of two-way active coupling between an operational atmospheric model and a wave model. What is novel and worthy of publication is the WASP parameterisation. For this reason, the actual expression of A and B would be a very good addition to the paper. However, the authors need to explain a bit more their choice of the WASP parameterisation, rather than using the Charnock values that WW3 can produce. A comparison of the WW3 Charnock and the WASP counterpart will be required and any major differences would need to be justified.

Those are good remarks and have also been pointed out by the Referee#1 (see pages 1 and 2).
Indeed, we did not use the classical approach that consists in directly taking the WW3 Charnock parameter, as there is a small variability in the Charnock parameter due to the wave field variability at a given wind speed and thus reducing the benefit of coupling with a wave model (see Fig. A).
As already mentioned, the WASP approach used here has two advantages: i) it allows to compare more directly the wave parameter coupled fields with observations, and so to check their validity; ii) the Charnock parameter is defined differently depending on the wind speed range considered, enabling to represent in a more physical way its behaviour and possible dependency on the waves. Especially, it reproduces the observed decrease of the drag coefficient by very strong wind, which would not be possible using a wave-age only Charnock parameter as in Drennan et al. (2005). The WASP parameterisation, unlike those based on wave-age Charnock parameters, is then usable for very strong wind speeds.
Section 2.3.2 has been enlarged to describe the advantages of using the WASP parameterization and an appendix has been added to the paper with the details of the calculation of the Charnock parameter (as shown above).

Anemometers mounted on buoys are rarely at 10m height. Nothing is mentioned regarding the adjustment of the buoy winds to 10m. The discussion regarding the bias reduction of 10m winds is only relevant if the buoy winds have been adjusted to 10m. Revise the manuscript accordingly.
Indeed, the observed wind speed and direction at Lion and Azur buoys and presented in our results are obtained after an adjustment to 10 meters using a standard log profile function.

The peak wave period is not a very stable quantity in situation of multi peak wave spectra. In the Mediterranean Sea, it is not too often the case, but over open ocean conditions, this is more often the norm. How would the WASP parameterisation deal with such situation? Is Tp computed from the full 2D-spectrum? Should one instead only determine Tp from the windsea part of the spectrum?
This is a very good remark, and the following details are now included in the sections 2.3.3 and 2.4 of the paper.
Indeed, the wind-wave coupling concerns the wind sea part of the wave field, and the peak period used to compute the Charnock parameter in WASP is actually the peak period of the wind sea only, as permitted by WW3 which has been modified (08-2017, Version 5.16) specifically to issue this variable as a coupling parameter.

Obviously, this paper is only a one case study. It has focussed on short range forecasts. This needs to be clearly highlighted and discussed.
In Janssen (2004), the impact of the coupling to waves is shown to be even more important at longer lead time. In the final section, it is discussed that ocean waves have an impact on the

momentum flux across the air-sea interface. However, according to Janssen and Bidlot (2018), waves might also impact on the latent and sensible heat fluxes https://doi.org/10.1016/j.piutam.2018.03.003
https://www.sciencedirect.com/science/article/pii/S2210983818300038

This is a good remark from both reviewers.
In the conclusion, we now discuss more the fact that the results are valid for our studied case, for short-range forecast in a deterministic framework and that other HPEs (and even other kinds of meteorological situations) are needed to further conclude on the sea state impact on high-resolution weather forecast.
Nevertheless, this study marks a new step in our understanding of the sea state impact on HPE, after Thevenot et al. 2016 and Bouin et al. 2017. Compared to these two previous studies, the case of october 2016 was particularly interesting due to the very strong wind regime at low-level generating a wind sea and very large heat and moisture fluxes below the easterly wind.
In fact, with this case study, we confirm that the dynamics of the system is really sensitive to sea state conditions but that very likely that the sea surface heat fluxes feeding HPEs are not.
Also, here we use an interactive atmosphere-wave coupling.

**Minor corrections:**

→ At a few places: biais -> bias
→ P2, line 26: waves, known as sea spray, occurs -> waves occurs, generating sea spray
→ P3, line 2: add Janssen 2004
→ P4, line 9: the adjustments of Bidlot et al. are only relevant if you use ST3, otherwise with ST4, there is an all new prescription of the whitecap dissipation that does not use Bidlot et al.
→ P5, line 28: the Charnock parameter is -> the surface roughness is
→ P8, line 7: well represents -> represents well
→ P8, line 8: It also can be noticed a delay -> Also, there is a delay
→ P14, line 18: In this purpose -> For this purpose
→ P14, line 21: coupled way -> coupled mode
→ P14, line 30: affected all the event – affected during all the event
All the corrections listed above have been included in the manuscript.

→ P5, in (7), the first term is the Charnock relation, the second term is the viscous contribution to z0. See Beljaars, A. C. M. (1994). The parametrization of surface fuxes in large-scale models under free convection. Q. J. R. Meteorol. Soc., 121, 255{270.
The sentence in the paper has been changed accordingly.

→ P6, line 15: to an untrained person, the relation between Tp and Ua would appear to be not correct. Truly speaking, one can find a relation between Tp ∼ c * Ua/g, with c non dimensional and from empirical fetch relation find that for a typical non dimensional fetch c ∼ 5, hence why one can simply write Tp ∼ 0.5 Ua
The direct relationship between Tp and Ua used by default in WASP (when no wave information is brought) has been more detailed.

additionnal refs:

Drennan et al. 2005, doi.org/10.1175/JPO2704.1

Rainaud et al. 2017, doi.org/10.1002/qj.3098

[revised manuscript text omitted]